

# Environment aware greedy deployment strategy for underwater acoustic sensor networks using bathymetric mapping and transmission loss modeling

Shekhar Tyagi, Akshat Shah and Abhishek Srivastava

Computer Science and Engineering, Indian Institute of Technology Indore, Indore, Madhya Pradesh, India

## ABSTRACT

Underwater acoustic sensor networks (UASNs) are rapidly evolving and serve a wide array of applications, including marine biology, underwater surveillance, oceanographic data collection, and disaster prevention. Despite notable technological advancements, UASNs continue to face critical challenges, including high propagation delays, limited bandwidth, and significant signal attenuation. To address these issues, this article proposes a novel greedy approach for optimal sensor deployment that aims to maximize coverage, minimize the number of sensors, and effectively account for transmission loss in underwater environments. The methodology begins with the extraction of a region of interest (RoI) using satellite imagery obtained *via* Google Earth. Following that is the construction of a bathymetry map that captures key topographical features. Recognizing the dynamic and complex nature of underwater environments, multiple simulation scenarios were developed to estimate transmission losses influenced by factors such as salinity, dissolved minerals (*e.g.*, magnesium and other salts), turbidity, pH, and temperature. These simulations help to evaluate how such environmental constraints affect acoustic propagation ranges and coverage estimations. The proposed approach was validated on a real-world RoI and benchmarked against existing methods, demonstrating up to a 10% improvement in coverage while requiring fewer sensors compared to traditional deployment techniques. Additionally, a prototypical real-world implementation was conducted to determine the optimal number of sensors required and the achieved coverage, thereby confirming the practicality and effectiveness of the method.

## INTRODUCTION

Beneath the mysterious depths of the ocean and other water bodies lies a world filled with both challenges and opportunities. This hidden environment is increasingly being explored with the help of underwater acoustic sensor networks (UASNs), an advanced smart technology designed to operate in conditions where conventional surface sensors cannot function. These networks are composed of small yet intelligent devices, known as

Corresponding author
Shekhar Tyagi,
shekhartyagicse@gmail.com

sensor nodes or motes, that are capable of detecting and measuring various underwater parameters such as temperature, chemical composition and marine life activity (*Heidemann, Stojanovic & Zorzi, 2012*).

Underwater wireless sensor networks (UWSNs), particularly UASNs, operate in harsh underwater environments where radio waves do not propagate effectively. These networks rely on sound waves for data transmission. Each sensor node is equipped with robust acoustic communication systems, energy-harvesting power sources, and compact processors capable of performing real-time data analysis.

These nodes form a flexible and interconnected network that can cover vast areas of the ocean. They are capable of tracking marine animals, detecting early signs of earthquakes or tsunamis, and supporting a variety of other applications. Using the unique properties of sound propagation in water, UASNs enable reliable data transmission even in deep, murky, or high-salinity environments where traditional systems often fail.

UASNs have numerous applications (*Akyildiz, Pompili & Melodia, 2005*) and are widely used in various underwater domains. Some key applications include monitoring water quality, measuring temperature and salinity levels, and detecting pollution, all of which help in identifying environmental changes and supporting the preservation of marine ecosystems. Military surveillance is another vital application that utilizes UASNs to strengthen the national security and defense capabilities by detecting submarines, underwater intruders, underwater mines and other unauthorized activities that may take place in underwater scenario. In seismic monitoring, such networks can help to detect underwater earthquakes and tsunamis by capturing seismic wave data, which can prove very vital for early warning systems and disaster preparedness, so that necessary steps can be taken before such disaster. The oil and gas industry uses UASNs for pipeline monitoring, leak detection, and thus ensuring safety and security. Oceanographic studies use UASNs for the collection of data of oceanic currents, marine habitats and underwater geology, essential for scientific research and understanding marine biodiversity.

Although, UASNs have various applications, they face several significant challenges. One major concern is the effect on communication range and bandwidth of the transmitted signal due to the high absorption and scattering of acoustic signals in water. Various salts like magnesium and boron in water leads to absorption losses, in addition to factors such as temperature, depth, pH, salinity and turbidity, all contributing to transmission losses. These factors collectively affect the data transmission range, signal speed, and network reliability. The underwater environment is dynamic in nature; so, due to water currents and variable temperatures it is more difficult to maintain the performance and stability of the network. To overcome these challenges an effective and optimal deployment of UASNs with an effective transmission loss model is needed.

Several studies have focused on the underwater deployment of sensors. *Xia et al. (2023b)* have introduced a hunting-style deployment model that enables real-time monitoring of changes in specific regions to track underwater pollution, and for this, they proposed a deployment method and an optimization model to identify optimal monitoring points in a 3-D underwater environment. *Senel, Akkaya & Yilmaz (2013)* have developed a novel approach to remotely deploy sensors to ensure maximum coverage and connectivity. The

data were collected from a surface sink node in UASNs. *Jin et al. (2018)* have presented a deployment optimization mechanism that utilizes depth-adjustable nodes (DODA) and aims to enhance coverage and maintain connectivity across all nodes. *Du, Xia & Zheng (2014)* designed an algorithm inspired by particle swarm optimization for underwater sensor networks, this aimed to detect dynamic event occurrences by optimizing sensor distribution. This algorithm was efficient in event coverage and fast convergence and suitability for distributed implementation. The Self-organized Proactive Routing Protocol for Non-Uniformly Deployed Underwater Networks (SPRINT) protocol (*Hyder et al., 2019*), a self-organized proactive routing scheme, eliminates the need for node localization and synchronization by relying on signal strength–based distance estimation. While it offers improvements in energy efficiency and throughput, it does not account for environmental factors such as salinity or temperature in its deployment strategy. To address deployment-related void problems and energy efficiency, the Remotely Operated Vehicle for Oceanographic and Bathymetric Applications (REOVA) protocol (*Khan, Aamir & Otero, 2024*) introduces a depth-aware strategic deployment scheme that selects cluster heads based on residual energy, link quality, and node density. However, its deployment logic primarily focuses on routing reliability, with limited environmental modeling. Another recent contribution proposes a grid-based depth-adjustable deployment method (DDOGDA) (*Aljughaiman, 2023*), incorporating GIS data and realistic underwater conditions for tsunami monitoring. The results show notable improvements in packet delivery ratio (by 22%), end-to-end delay (by 266%), and energy consumption (by 183%) over baseline strategies. However, it does not explicitly model acoustic signal losses due to environmental parameters like pH or turbidity. Furthermore, a power-control–based deployment approach (*Wen et al., 2024*) considers asymmetric links and integrates a novel crow-colony search optimization algorithm (C-CSOA) to reduce power usage. This method achieves nearly 24% reduction in power consumption but does not address bathymetric variations in real-world aquatic terrains. Collectively, while these recent methods provide valuable advances in routing efficiency, strategic deployment, and power control, they often overlook fine-grained environmental modeling and realistic terrain-driven acoustic propagation analysis—gaps that our proposed method addresses through the integration of bathymetric mapping and multi-parameter signal loss estimation.

These contributions have helped improve the capabilities of UASNs by proposing various deployment strategies and optimization techniques to efficiently utilize their wide range of applications. However, these advances have few shortcomings and research gaps. Firstly, maximal coverage claimed by these methods often comes at a cost of very high computational complexity. Secondly, none of these researches addresses the effects of transmission losses or other underwater environmental conditions. There is a lack of research on how these losses vary with depth and their impact on the optimal propagation range of the sensors. In most of the previous studies it has also been assumed that all sensors behave ideally in all types of underwater environments regardless of waves motion, currents, salinity, turbidity, water temperature, depth and other environmental constraints.

In this article, a robust deployment algorithm, along with coverage calculation, is proposed that incorporates the effect of factors such as absorption losses and transmission losses influenced by depth, salinity, water temperature and pH level. Our contributions are as follows:

1. Designing bathymetry to gain insights of the topography of the region of interest (RoI) for accurate analysis and optimizing the deployment.
2. Construction of a novel greedy algorithm—a heuristic approach in which the next best sensor location is selected based on local optimization at each step, without considering the global optimum—that systematically incorporates underwater environmental conditions to optimize sensor placement.
3. Integration of continuous coverage calculation model upon each sensor placement to maximize coverage efficiency, finally crafting a streamlined approach for optimal sensor deployment and coverage calculation, with reduced computational complexity and high accuracy rate.

The remainder of this article is organized as follows: 'Related Work' discusses related work in this field; 'Proposed Methodology' details the proposed methodology; 'Results and Evaluation' describes the experiments and results validating the efficacy of the proposed approach through comparisons with existing methods followed with a real-world prototype implementation. Finally, 'Conclusion and Future Work' concludes the article.

# RELATED WORK

Based on previous studies in this domain, the related work can be categorized into two main node deployment strategies: volume coverage-based and target coverage-based approaches. Volumetric coverage-based node deployment focuses on ensuring that sensor nodes are strategically placed to maximize coverage over a given volume or area. The target coverage-based deployment method aims to deploy nodes specifically to monitor or track some specific targets within the network's coverage area.

## Volumetric coverage-based node deployment

*Jin et al. (2018)* implemented a node optimization technique that was based on adjusting node depths, so that redundant coverage areas can be minimized with neighboring nodes, improving overall network coverage. *Wang & Wang (2017)* developed a depth determination method using breadth-first or depth-first search algorithms to assess network connectivity. They utilized Voronoi diagrams to identify coverage gaps among underwater nodes and applied the K-means clustering algorithm to adjust node depths on the water surface. *Akkaya & Newell (2009)* introduced a fully distributed technique for self-reconfiguration of UWSNs that focuses on adjusting the sensor node depths after deployment to optimize initial coverage. *Su, Fan & Fu (2020)* proposed a Voronoi-based optimal depth adjustment deployment scheme where gateway nodes gather sensor node coordinates to construct Voronoi diagrams. Leader nodes are strategically placed on the water surface with the help of of this diagram, while other nodes are adjusted to varying depths to reduce coverage overlaps and enhance network efficiency. *Alam & Haas (2008)*

proposed a methodology in which they suggest that the ratio of the communication range and the sensing range can help in placing node in each cell created by assuming the 3-D space as a truncated octahedron. *Xiaoyu, Lijuan & Linfeng (2013)* implemented their approach by randomly deploying the sensors in the 3-D water environment and analyzing the complexity and coverage optimization and comparing it with the deployment in 2-D plane. *Cobanlar et al. (2022)* developed an optimization framework to model the impact of parameter k on the lifetime of UASNs for a balance between k-connectivity and network longevity. Lastly, *Xia et al. (2023a)* implemented an optimization algorithm that was inspired with the activities of moth around flame, they incoporated fuzzy operators to improve network coverage and maintaining continuous connectivity along with minimal node deployment. The fuzzy operators dealt with rectifying uncertainties and inaccuracies in broadcasting data which is quite a common issue in underwater scenarios. Their methodology focus on optimal resource utilization and enhancing efficiency of UASNs.

## Target coverage based node deployment

*Bharamagoudra & Manvi (2016)* proposed a scheme for deploying 3-D UASN architectures using static sensor nodes. Their focus was optimal node positioning and route planning through mathematical modeling to support their applications. *Arivudainambi, Balaji & Poorani (2017)* used the cuckoo search algorithm to optimize node placement which aims for coverage of specific underwater targets. Their approach enhances node deployment efficiency by ensuring target coverage without disturbing network connectivity. *Jiang, Feng & Wu (2016)* developed a redeployment algorithm for UASNs inspired by the predatory behavior of wolves and integrated crowd control mechanisms to ensure comprehensive event coverage and avoid local optimization traps. *Xia et al. (2023b)* designed a model for real-time monitoring of specific regions, they named this deployment model as hunting style technique, as it was specific for event monitoring, that aimed to track underwater pollution. They optimally deployed sensors over a region in 3-D underwater environments for effective monitoring. *He et al. (2017)* in their methodology designed a simulator termed as fish simulator, which was inspired by the fish moments over an underwater target event. Based on this, they successfully adjusted the location of sensors and coverage rate of UASNs. The fish simulation behaviors include schooling and swarming, which allows the sensors to dynamically adjust their position and enhance coverage. This technique ensured that the sensor nodes adapt to variable conditions and can maintain optimal coverage and enhance the ability of the network. *Dong et al. (2023)* implemented an energy efficient approach in their methodology for optimal coverage of target events using the deployed sensors. In this methodology careful planning of sensor deployment was done and operation schedules were made to ensure that the network keeps working efficiently for extended periods. These studies showcase a variety of innovative approaches and methodologies that address the challenges faced by UASNs.

## PROPOSED METHODOLOGY

This section presents the proposed approach for sensor deployment in UASNs, with the objective of optimizing coverage while accounting for the dynamic characteristics of

underwater environments. A key challenge in deploying UASNs is the difficulty in accurately understanding the topography of the RoI, such as oceans, lakes, or rivers. To address this, the proposed methodology utilizes bathymetric maps, which represent the depth and structure of the underwater environment, including the seafloor and other submerged features. In addition to topographical challenges, the approach also considers critical environmental constraints such as signal attenuation, absorption, and transmission losses. These are modeled using established underwater acoustic loss models, which take into account variables such as salinity, pH, depth, turbidity, and temperature. The computed losses are used to estimate the effective propagation range of each sensor, allowing for more accurate coverage calculations.

Furthermore, the methodology incorporates a greedy algorithm-based coverage estimation technique to ensure optimal sensor placement. This algorithm iteratively selects deployment locations that maximize incremental coverage, aiming to achieve full coverage with the minimum number of sensors required. By integrating bathymetric analysis with environmental constraint modeling and a strategic deployment algorithm, the proposed method addresses the core challenges in UASN deployment, and enhances network efficiency and performance.

The flow of the methodology is illustrated in the Fig. 1. The complete methodology will be detailed in the subsequent sections that follow.

## Data acquisition–RoI extraction of a water body (pond, lake, ocean) using Google Earth

Google Earth (Google, 2024) provides a functionality known as 'Add Path or Polygon' to select points on an area. For the extraction of an RoI, firstly the geographical coordinates are chosen and multiple points which are close to each other (on google earth panel) are selected, making a path leading to a polygon shape over a water body, ensuring that maximum content of the RoI is covered with these points. This collection of points represent the RoI and Google Earth also provides a feature to save the selected points into a keyhole markup language (KML) file.

The process of extracting an RoI is shown in Fig. 2. The KML file consists of latitude and longitude information of all the selected points. However, the KML file does not contain the depth information which is required for further processing. In order to gather depth data using the coordinates (latitude and longitude), an online converter (*Advanced Converter, 2024*) is utilized. The Advanced Converter retrieves bathymetric depth data using global datasets such as *GEBCO Compilation Group (2023)* and SRTM30 PLUS *SRTM Plus Compilation Group (2009),* based on the provided coordinates. The typical resolution of this data ranges from 15 to 30 arc-seconds (approximately 500 m to 1 km), which is sufficient for medium-scale bathymetric mapping and sensor deployment planning.

A sample representation of depth retrieval using coordinates is shown in Fig. 3. All the gathered data (latitude in degrees, longitude in degrees, and depth in meters) is stored as a comma separated values (CSV) file for further processing. A sample CSV file is shown in Fig. 4.

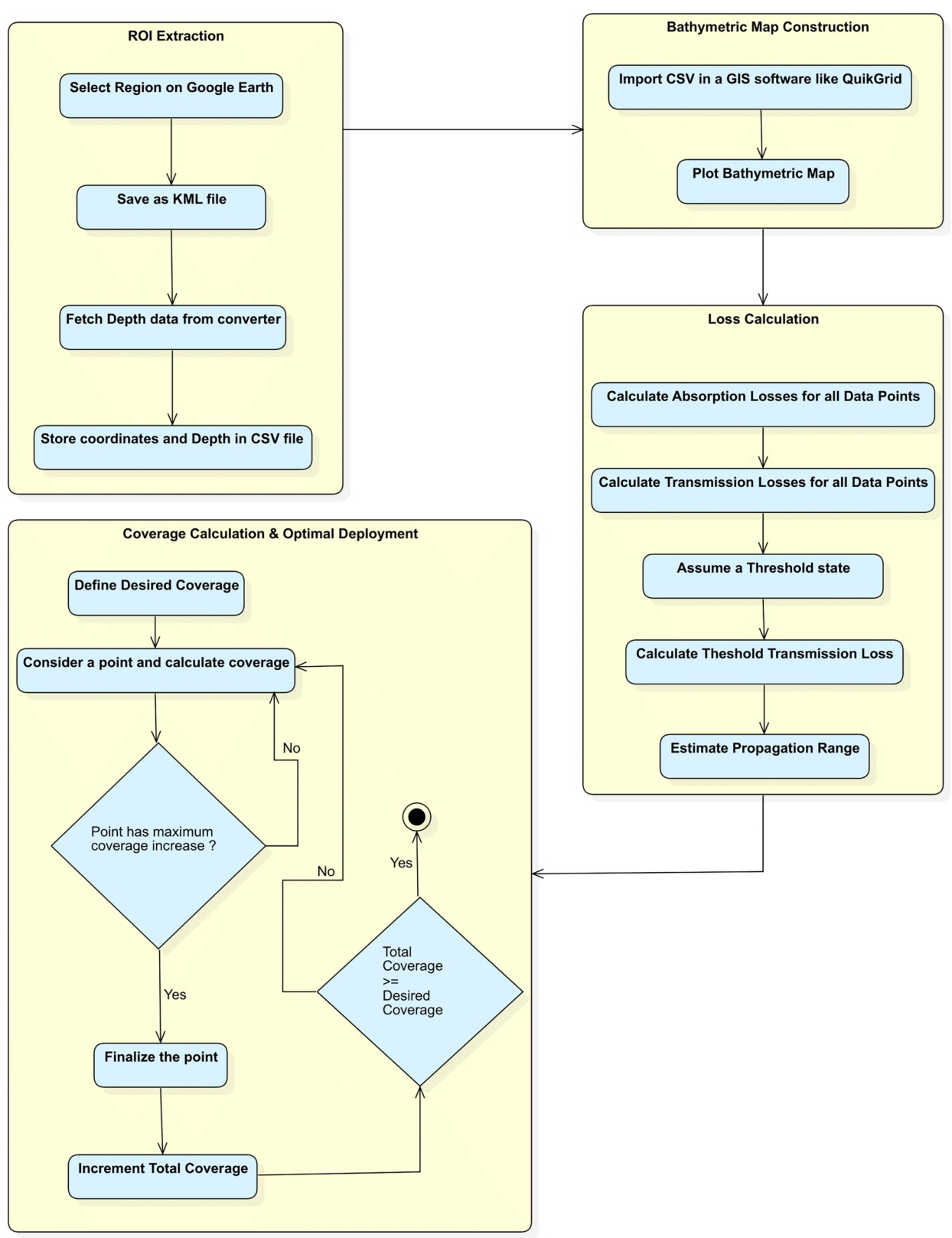

**Figure 1 Flow of the proposed methodology.**

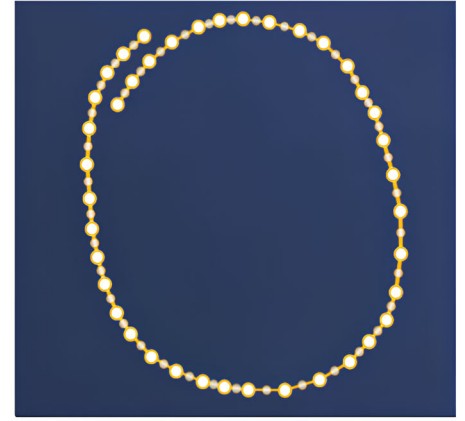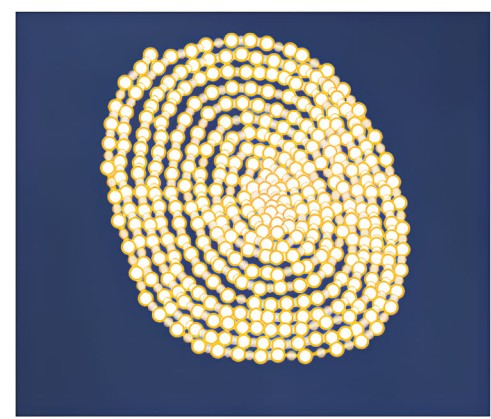

**Figure 2 Extracting the region of interest (RoI).**

**Popular Converters**

»» Length Converter
»» Mass(Weight) Converter
»» Volume Converter
»» Age Calculator
»» Download Time Calculator
»» Color Converter
»» Fuel Consumption Converter
»» Your Age in Future
»» Sunrise-sunset time
»» Fuel Cost Calculator

**Map tools**

»» Find distance between places
»» Area-perimeter calculator
»» Distance of a path
»» Find coordinates of a location
»» Elevation on map
»» Postal code
»» Address finder

**Altitude 19.0857512,85.7254165**

Find altitude by coordinates on google maps in meters and feet. Find altitude **filling in input fields (latitude, longitude)** or **doing click on the map**. Drag to change location and find new elevation.

**Altitude: -1587 m. (-5207 ft)**               [          ] [          ] [Search]

Full screen

Leaflet | © OpenStreetMap contributors

**Figure 3 Depth retrieval.**         

| | A | B | C |
|---|---|---|---|
| 1 | Latitude | Longitude | Depth |
| 2 | 19.09211 | 85.71556 | -1535 |
| 3 | 19.09184 | 85.71427 | -1533 |
| 4 | 19.09148 | 85.71326 | -1530 |
| 5 | 19.0908 | 85.71225 | -1529 |
| 6 | 19.09013 | 85.71148 | -1529 |
| 7 | 19.08906 | 85.71089 | -1531 |
| 8 | 19.08796 | 85.71049 | -1533 |
| 9 | 19.08692 | 85.71036 | -1536 |
| 10 | 19.08595 | 85.71019 | -1538 |
| 11 | 19.08499 | 85.71019 | -1542 |

**Figure 4** **A sample CSV file with latitude in degrees, longitude in degrees, and depth in meters.**

## Construction of bathymetric map of the extracted RoI

Bathymetric maps (*Smith, Sandwell & Raney, 2005*) are specialized maps that depict the topography and physiographic features of ocean, sea bottoms, lakes, river beds and other such water bodies. These maps are able to represent the depth and shape of underwater features in 2-Dimensions, making it easier to further analyze the topography in detail. Bathymetric maps are widely utilized to study effects of climate change, monitor sea level rise, beach erosion, coastal flooding and subsidence (land sinking). The bathymetric data provides a basis to create models that can calculate water temperature, salinity, tides and currents in an area and assist in conservation and monitoring of the underwater flora and fauna (*National Ocean Service, 2024*). Bathymetric maps can be created using traditional survey methods, echo sounding, SONAR, underwater vehicles such as autonomous underwater vehicles (AUVs) and remotely operated vehicles (ROVs), or satellite altimetry. In our approach, bathymetric maps are created using a CSV file obtained after extracting the RoI from Google Earth. This CSV file is then processed using QuikGrid (*Coulthard, 2007*), a specialized software tool, to construct a two-dimensional bathymetric map of the RoI. To ensure accurate topographical modeling, bathymetric data was extracted using an online converter tool (*Advanced Converter, 2024*) linked to global elevation datasets such

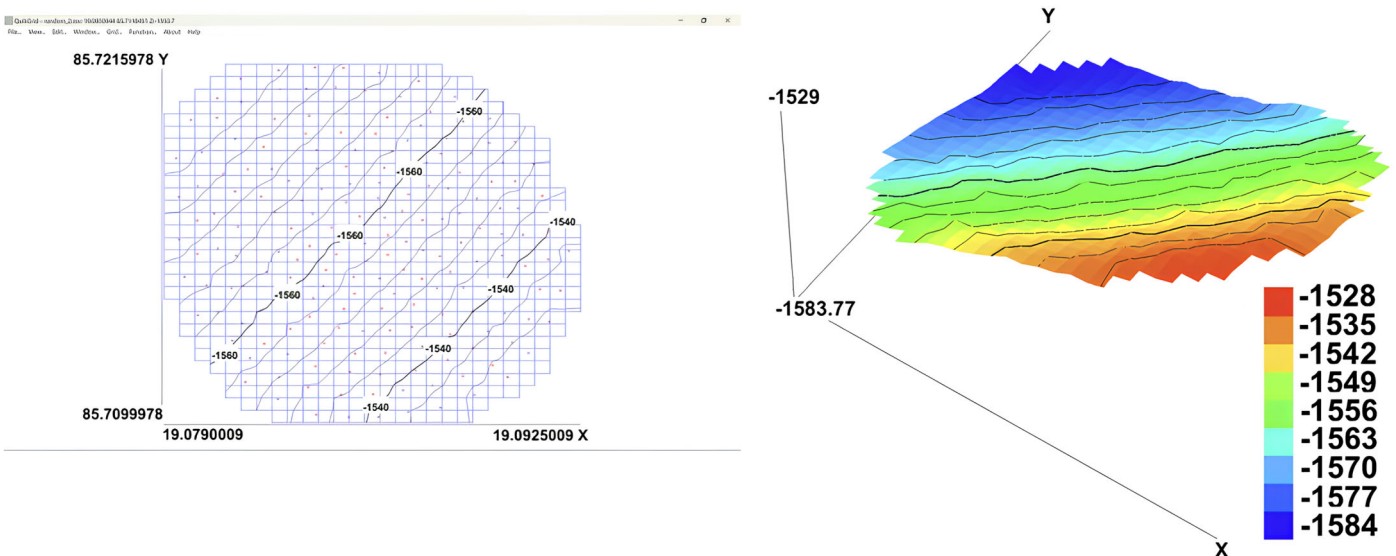

**Figure 5 2D and 3D bathymetric maps of the extracted region of interest (RoI).** In both maps, longitude (in degrees) is represented along the Y-axis and latitude (in degrees) along the X-axis, while in the 3D map, depth (in meters) is visualized along the Z-axis.

as SRTM and General Bathymetric Chart of the Oceans (GEBCO). While these sources may exhibit resolution limitations—typically ranging from 3 to 10 m spatially and vertical accuracy between ±1 to ±3 m—their error margins are relatively small in the context of large-scale underwater deployments. Although minor inaccuracies in depth profiling may slightly affect acoustic modeling or range estimation, they do not significantly alter the overall simulation outcomes. Thus, the extracted bathymetric data provides a sufficiently accurate and dependable representation of underwater terrain for deployment and coverage optimization.

Figure 5 shows the 2-Dimensional and 3-Dimensional view of the Bathymetric map of extracted RoI.

## Computing absorption and transmission losses

Though acoustic propagation is one of the most appropriate methods for underwater monitoring, it faces several challenges like absorption and transmission losses. In UASNs, attenuation contribute to various losses resulting in reduction in intensity of sound as it travels through water.

*Absorption loss* wave currents and presence of salts like magnesium and boron leads to absorption of sound energy in the water, this reduction in sound intensity is termed as absorption loss.

*Transmission loss* refers to the reduction in signal strength amid various factors such as absorption, salinity, turbidity, depth, water temperature, *etc*. So, transmission loss can termed as a comprehensive loss of signal intensity incorporating the absorption loss and other addon factors.

The Ainslie and McColm model (*Ainslie & McColm, 1998*) provided several conclusions in their transmission loss model, which are incorporated in our proposed approach:

- Decreasing pH, decreases low frequency absorption up to $f_1$ with no effect at higher frequencies.
- Increasing salinity will lead to a decrease in absorption at low frequency ($\leq f_1$) and increases absorption at high frequency ($\geq f_1$).
- Increasing temperature leads to a decrease in absorption at all frequencies, except in the immediate proximity of the relaxation frequencies $f_1$ and $f_2$, where absorption is increased.
- Increasing depth will lead to decrease in absorption at high frequency ($\geq f_1$), with particularly no effect at the lower frequencies.
- Turbulence caused because of the wave activities, increases absorption and transmission loss across the frequency spectrum.

The model for computing the relaxation frequencies (in KHz) amid the presence of boron and magnesium salts in underwater environment is mathematically given as:

$$f_1 = 0.78 \left(\frac{S}{35}\right)^{1/2} e^{T/26} \quad \text{(for boron salts)}, \tag{1}$$

$$f_2 = 42 e^{T/17} \quad \text{(for magnesium salts)}. \tag{2}$$

Here, $S$ specifies the salinity in parts per thousand, $T$ is temperature in °C.

Absorption loss (*Ainslie & McColm, 1998*; *Tsuchiya, 2010*) is computed as follows:

$$\alpha = 0.106 \frac{f_1 f^2}{f_1^2 + f^2} e^{\frac{pH-8}{0.56}} + 0.52 \left(1 + \frac{T}{43}\right) \left(\frac{S}{35}\right) \frac{f_2 f^2}{f_2^2 + f^2} e^{-\frac{z}{6}} + 0.00049 f^2 e^{-\left(\frac{T}{27} + \frac{z}{17}\right)} \tag{3}$$

where $\alpha$ is the absorption coefficient (in dB/km), $z$ is depth (in km), and $f$ is frequency (in kHz).

Transmission loss (*Tsuchiya, 2010*) is computed as:

$$TL = 10k \log_{10} r + \alpha r 10^{-3} \tag{4}$$

where $TL$ is the transmission loss (in dB), r is the propagation range (Kms), $k$ is the spreading factor, ($k = 1$ for cylindrical spreading and $k = 2$ for spherical spreading).

## Estimating the propagation range of sensor, with varying losses and depth scenarios over the extracted RoI

Amid the presence of various losses, sensors may not be able to transmit at their full potential. This can lead to a reduction in their maximum effective transmission range, which may vary under different conditions, as discussed in the preceding section.

For estimating the propagation range of sensors in water, our methodology first computes a threshold transmission loss TL_TH, a defined environmental condition at which sensor communication is just feasible—based on parameters such as frequency, salinity, temperature, depth, pH, and ideal propagation range, using Eq. (4). The value of

TL_TH is the transmission loss value at the point present at the maximum depth because transmission loss is least at maximum depth (*Lunkov & Shermeneva, 2019*). Therefore, the value of TL_TH is set depending on the maximum depth of the RoI which may vary for different underwater regions. This TL_TH represents the maximum acceptable transmission loss, which means all the data points above the threshold state have transmission loss value greater than TL_TH. Finally, all those points then will undergo a reduction in propagation range as per the calculative assessments defined in Eqs. (5) and (6).

The approach calculates the transmission losses $TL_i$ for each data point in the RoI, assuming $r_i$ as the ideal propagation range, using Eq. (4):

$$TL_i = 10k\log_{10}r_i + \alpha r_i 10^{-3}. \tag{5}$$

If $TL_i > TL\_TH$:

- This means the propagation range needs to be reduced. The adjusted $TL'_i$ is calculated using:

$$TL'_i = TL\_TH - \delta_i, \quad \text{where } \delta_i = TL_i - TL\_TH. \tag{6}$$

If $TL_i \leq TL\_TH$:

- This means there is no need to reduce the propagation range *i.e.*, no adjustment is required, and $TL_i' = TL\_TH$

Now, by using the adjusted transmission loss $TL_i'$, our methodology progresses to compute the new propagation range $r_i'$ for each data point. This is accomplished by applying Eq. (4) with the adjusted transmission loss values, taking into account the varying environmental conditions across different depths and locations within the RoI.

## Optimal deployment and coverage calculations

Sensor deployment in underwater environments is a complex task that requires strategic placement to ensure successful data transmission and maximize coverage of the RoI with minimal sensors.

The optimality of sensor deployment depends on carefully defining and extracting the RoI. It must be well-defined to avoid gaps that could create uncovered areas. This ensures effective sensor network deployment, covering the entire RoI without significant blind spots.

*Coverage* is defined as the proportion of covered data points within the range of a sensor relative to the total number of data points, and is mathematically given as:

$$C = \frac{\sum_{i=1}^{N_c} x_i}{N_t} \tag{7}$$

where:

- $C$ is the coverage.
- $N_c$ is the number of covered data points.
- $N_t$ is the total number of data points.
- $x_i$ represents individual covered data points.

The methodology for sensor placement and coverage calculation is designed to maximize coverage within an RoI. The steps are as follows: First a desired coverage level is defined, ranging from 0 to 1, where 1 represents 100 percent coverage. This target coverage level is the goal of the algorithm for strategic placement of sensors. In the next step, an initial potential sensor position is selected at a point $P_i$ from the available data points within the RoI. To calculate the sensor's potential coverage area, the Euclidean distance from $P_i$ to all other points in the RoI is calculated. This distance helps to determine the proportion of RoI that would be covered if a sensor was placed at $P_i$. To determine the effectiveness of placing a sensor at $P_i$, the algorithm calculates the increase in coverage that $P_i$ would provide. Then it evaluates the proportion of the RoI covered by $P_i$ and compares this increase with the potential increases from other sensor locations. If placing the sensor at $P_i$ gives a maximum coverage increase compared to other locations, the sensor position at $P_i$ is finalized. If not, the algorithm skips $P_i$ and evaluates the next potential sensor position and repeats the coverage calculation process. Once the optimal sensor position is identified, the total coverage is incremented by adding the coverage provided by the sensor at $P_i$. After this, the algorithm selects a new data point as the next potential sensor location and repeats the coverage calculation and comparison process. This iterative procedure evaluates potential sensor locations, calculates the coverage given by each location and compares the coverage increase to find the position that contributes highest to the overall coverage. The algorithm continues this iterative process of selecting potential sensor locations, calculating coverage, and comparing coverage increases until the total coverage of the RoI meets or exceeds the value of desired coverage. Through this systematic method, sensors are strategically placed to maximize coverage efficiently within the defined region, ensuring that the placement process achieves the desired coverage level using minimal number of sensors.

A sample working of the coverage estimation is shown in Fig. 6. The RoI contains some irregularities, indicating that the distribution of data points is not uniform. The goal is to achieve a coverage of 1.0.

- Total number of data points: 36
- Desired coverage: 1.0

**Step 1: Finding the first sensor position**

The algorithm starts by finding the point from the set of data-points (black) that provides the maximum increase in coverage. This point is selected and shown in green.

**Step 2: Placing sensor 1**

In this step, sensor 1 denoted by yellow is placed at the determined position. The coverage of sensor 1 is shown in (A).

- Sensor 1 covers 6 out of 36 data points.
- Total coverage $= \frac{6}{36} = 0.167$

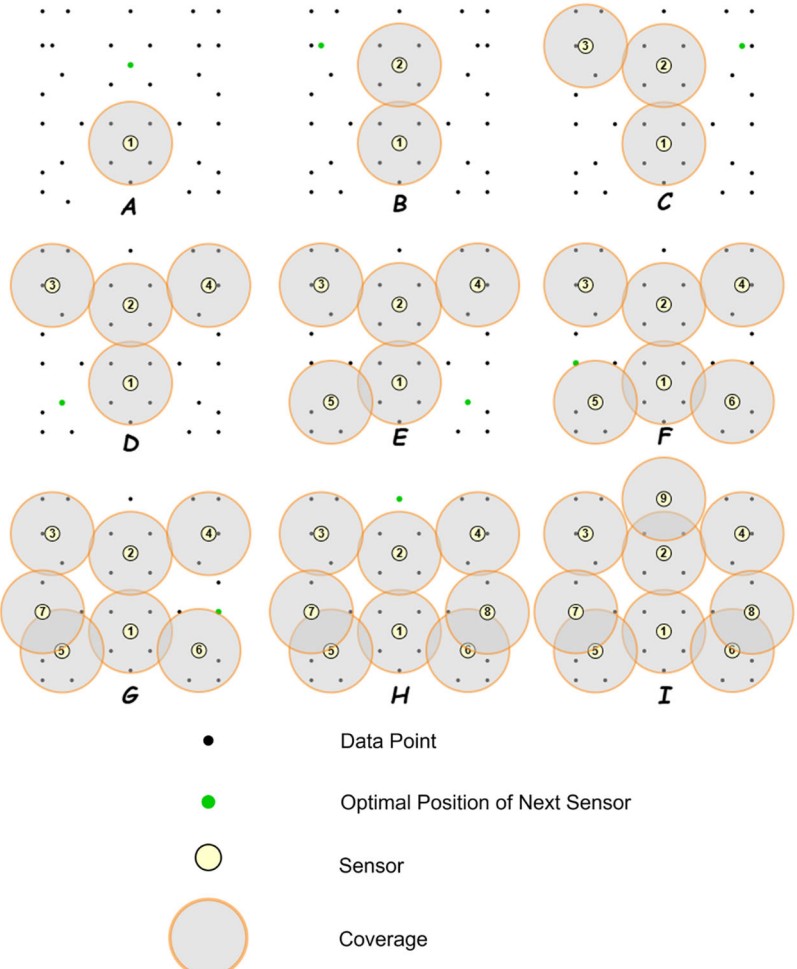

**Figure 6 A sample schematic working of the deployment algorithm with coverage calculation.**

Next, the algorithm finds the next point with the maximum coverage increase.

**Step 3: Placing sensor 2**

In this step, sensor 2 is placed at the next determined position. The updated coverage is shown in (B).

- Sensor 2 increases the total coverage to 11 out of 36 data points.
- Total coverage $= \frac{11}{36} = 0.3056$

Next, the algorithm finds the next point with the maximum coverage increase.

**Step 4: Placing sensor 3**

In this step, sensor 3 is placed at the next determined position. The updated coverage is shown in (C).

- Sensor 3 increases the total coverage to 16 out of 36 data points.

- Total coverage $= \frac{16}{36} = 0.44$

Next, the algorithm finds the next point with the maximum coverage increase.

**Step 5: Placing sensor 4**

In this step, sensor 4 is placed at the next determined position. The updated coverage is shown in (D).

- Sensor 4 increases the total coverage to 21 out of 36 data points.
- Total coverage $= \frac{21}{36} = 0.5833$

Next, the algorithm finds the next point with the maximum coverage increase.

**Step 6: Placing sensor 5**

In this step, sensor 5 is placed at the next determined position. The updated coverage is shown in (E).

- Sensor 5 increases the total coverage to 25 out of 36 data points.
- Total coverage $= \frac{25}{36} = 0.6944$

Next, the algorithm finds the next point with the maximum coverage increase.

**Step 7: Placing sensor 6**

In this step, sensor 6 is placed at the next determined position. The updated coverage is shown in (F).

- Sensor 6 increases the total coverage to 29 out of 36 data points.
- Total coverage $= \frac{29}{36} = 0.8056$

Next, the algorithm finds the next point with the maximum coverage increase.

**Step 8: Placing sensor 7**

In this step, sensor 7 is placed at the next determined position. The updated coverage is shown in (G).

- Sensor 7 increases the total coverage to 32 out of 36 data points.
- Total coverage $= \frac{32}{36} = 0.89$

Next, the algorithm finds the next point with the maximum coverage increase.

**Step 9: Placing sensor 8**

In this step, sensor 8 is placed at the next determined position. The updated coverage is shown in (H).

- Sensor 8 increases the total coverage to 35 out of 36 data points.
- Total coverage $= \frac{35}{36} = 0.9722$

Next, the algorithm finds the next point with the maximum coverage increase.

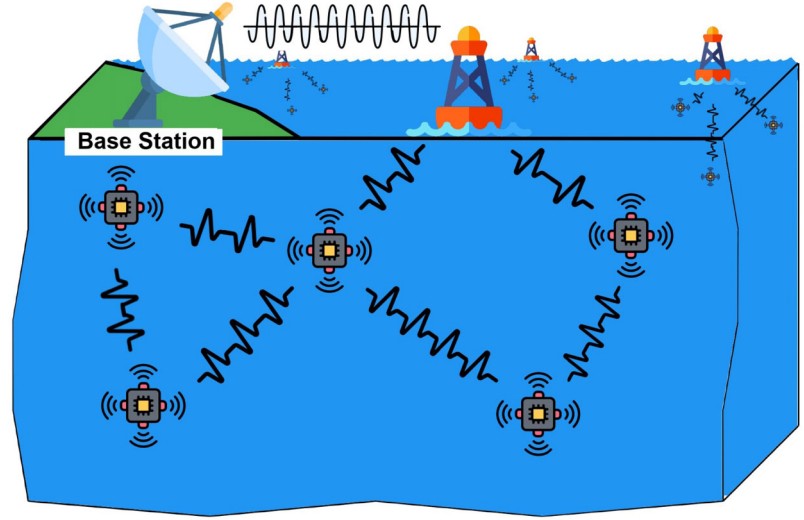

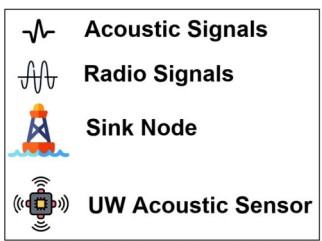

**Figure 7** **Illustration of the proposed sensor deployment architecture over an extracted RoI.** Multiple sensor nodes are placed using the Optimize Underwater Sensor Deployment Algorithm. Each group of sensors communicates acoustically with an assigned Sink Node, which then transmits the aggregated data *via* radio communication to the Base Station. The Base Station may be located onshore or on a floating platform. This layered communication strategy ensures reliable data transfer and full area coverage.

**Step 10: Placing sensor 9**

In this step, sensor 9 is placed at the next determined position. The updated coverage is shown in (I).

- Sensor 9 increases the total coverage to 36 out of 36 data points.
- Total coverage $= \frac{36}{36} = 1.0$

Since the total coverage of 1.0 is attained, the algorithm stops.

This step-by-step example demonstrates the process of sensor deployment where each sensor is placed at a position that maximizes the coverage increase and the process continues iteratively until the 100% desired coverage level is achieved. This methodology works in a 3-D scenario, considering spherical spreading for the proposed approach.

Figure 7 represents a sample setup for the deployment architecture of the proposed approach involving sensors in an extracted RoI. It shows multiple sensors placed using the Optimize Underwater Sensor Deployment Algorithm, communicating with each other. A sink node is assigned to each group of sensors in its range. This sink node collects data

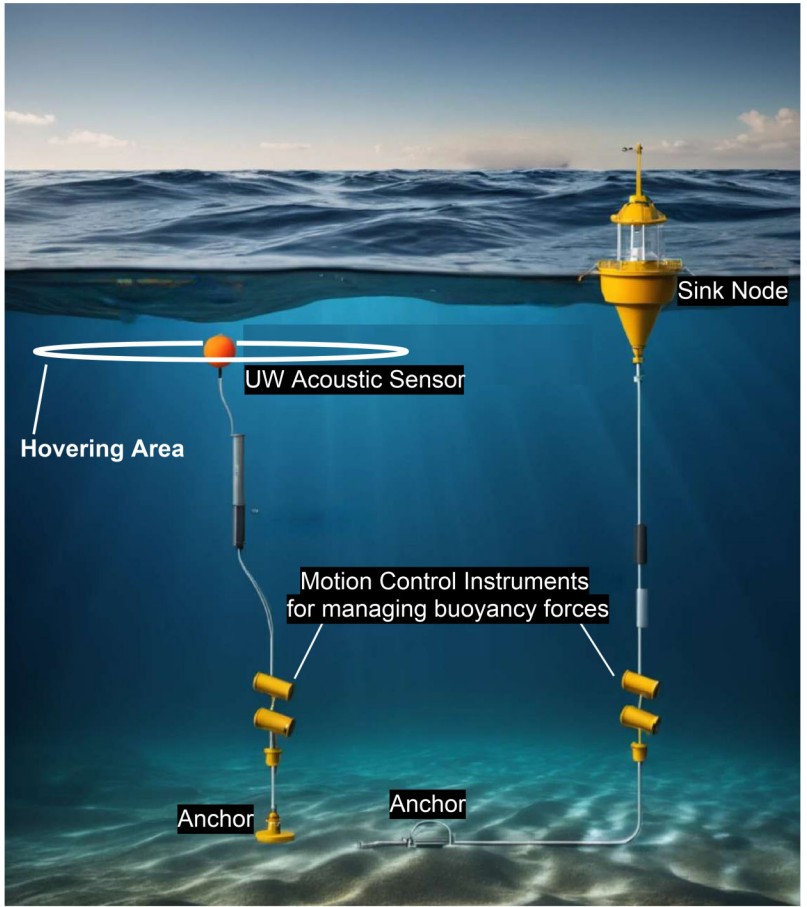

**Figure 8 Sample system setup showing anchored sensor and sink node stabilized using buoyancy control.**

from all the sensors in its range through acoustic communication (acoustic signals) and then transmits the information using radio communication (radio signals) to the base station. This base station can be located on land, or present on water surface on a ship. Multiple sink nodes are placed to collect data from different groups of sensors to ensure optimal connectivity and coverage. The information collected at the base station can then be used for monitoring purposes.

Figure 8 represents sample arrangement of the system showing the placement of a sensor and a sink node. The sensor floats in underwater within its hovering area through an anchor fixed under the seabed. The hovering area is the region beyond which sensor cannot reach while it is fixed with the anchor. The design and weight of anchor is responsible to reduce drifting of the sensor beyond its hovering range. Sink node is also fixed through an anchor to float on the surface within its own hovering range. Motion control instruments are also deployed above the anchors in order to handle the buoyancy forces acting on the sensor and the sink node. These instruments are equipped with small ballast tanks (*Tiwari & Sharma, 2021*) to avoid sinking or floating much away from the hovering range. The concepts of buoyancy and Archimedes principle ensures that the sensor and sink node, being connected

to respective anchors at the sea bed, do not displace too much form their positions. According to Archimedes Principle (*Kireš, 2007*), the buoyant force on an object is equal to the weight of fluid displaced by the submerged object. This force is calculated by:

$$F_b = \rho \cdot V \cdot g \tag{8}$$

where $F_b$ is the buoyant force, $\rho$ is the density of the fluid, $V$ is the volume of the object, and $g$ is the acceleration due to gravity.

The weight of the object (sensor or sink node) $W$ is known.

The net force $F_{\text{net}}$ is given by:

$$F_{\text{net}} = W - F_b. \tag{9}$$

- If $F_{\text{net}} > 0$, the object will tend to sink. So, the motion control instruments need to decrease the weight in order to avoid sinking.
- If $F_{\text{net}} < 0$, the object will tend to rise. So, the motion control instruments need to increase the weight in order to avoid rising.
- If $F_{\text{net}} = 0$, the object will remain in its position and there is no need for any adjustments.

This comprehensive deployment approach is stated mathematically in the Algorithm 1, aiming to strategically deploy sensors in underwater environments to achieve a desired coverage level efficiently. The strategy concludes by dynamically adjusting sensor ranges, evaluating potential deployment locations and incorporating environmental constraints. Thereby coverage is maximized and finally the coordinates of the deployed sensors are returned.

## Complexity analysis

The methodology aims to solve the problem of strategically placing sensors in a 3-D underwater space to maximize coverage. It begins with the pre-computation of Euclidean distances of each sensor node with all other grid points, which is achieved in $O(mn)$ time, where $m$ is the number of sensor nodes and $n$ is the number of grid points. It maintains a distance matrix for these computations.

The major part of the methodology is its iterative sensor placement loop, which repeats until full coverage is achieved or no further significant increase in coverage can be obtained. In each iteration, potential sensor positions are evaluated to determine whether they contribute the most to enhancing coverage. This maintains a priority queue to efficiently select the first sensor placement that provides the maximum coverage increase. The priority queue operations-insertion and extraction are of logarithmic complexity relative to the number of grid points $O(\log n)$.

Hence, the complexity for this sensor placement along with coverage calculation loop is $O(m^2 \log n)$. This complexity is calculated from the combination of $m$ iterations (one for each sensor placement) and the logarithmic complexity of priority queue operations for each iteration. At last, the complexity of updating the coverage and adjustments to remaining grid points is $O(n)$ operations per iteration in worst case. Therefore, the overall complexity of $O(m^2 \log n)$ is achieved.

---

**Algorithm 1** Optimize underwater sensor deployment.

1: **Input:** RoI points, Desired coverage, Environmental factors (Salinity, temperature, pH, depth)
2: **Output:** Deployed sensors coordinates
3: **Procedure** OPTIMIZE UNDERWATER SENSOR DEPLOYMENT (RoI_points, Desired_coverage, Environmental_factors)
4:        Coverage $\leftarrow$ 0, Deployed_sensors $\leftarrow$ []
5:        Compute TL_TH based on environmental factors
6:        **while** *Coverage < Desired_coverage* **do**
7:                Best_sensor_position $\leftarrow$ None, Max_coverage_increase $\leftarrow$ 0
8:                **for** each $P_i$ in RoI_points **do**
9:                        **if** *$P_i$ is covered by Deployed_sensors* **then**
10:                            **Continue**
11:                        **end if**
12:                    Calculate potential coverage_increase
13:                    Determine range $r_i$ and compute TL_i
14:                    **if** *TL_i > TL_TH* **then**
15:                            Adjust $r_i$ to maintain TL_TH
16:                    **end if**
17:                    **if** *coverage_increase > Max_coverage_increase* **then**
18:                            Update Best_sensor_position and Max_coverage_increase
19:                    **end if**
20:                **end for**
21:                **if** *Best_sensor_position is None* **then**
22:                        **Break**
23:                **end if**
24:                Deploy sensor at Best_sensor_position, update Coverage and RoI_points
25:        **end while**
26:        **Return** Deployed_sensors
27: **end procedure**
28: **function** COMPUTE TRANSMISSION LOSS (RoI_points, Environmental_factors)
29:        Initialize TL_matrix with zeros
30:        **for** each pair $(P_i, P_j)$ in RoI_points **do**
31:            Calculate range $r_{ij}$ and compute TL_ij
32:            Store TL_ij in TL_matrix
33:        **end for**
34:        **Return** TL_matrix
35: **end function**
36: **function** ADJUST RANGE FOR TLTH ($r_i$, $TL_i$, $TL\_TH$)
37:        $r_i$_adjusted $\leftarrow$ $r_i$
38:        **if** *$TL_i > TL\_TH$* **then**
39:            Adjust $r_i$_adjusted to achieve $TL\_TH$
40:        **end if**
41:        **Return** $r_i$_adjusted
42: **end function**

---

The time estimation model calculates the time taken for various phases in the entire proposed methodology. The total time taken is given by:

$$T_{\text{total}} = T_{\text{extract}} + T_{\text{bathymetry}} + T_{\text{loss,range}} + T_{\text{deployment}} \tag{10}$$

where, $T_{\text{extract}}$ is time taken to extract coordinates and depth data of the ROI, $T_{\text{bathymetry}}$ is the time taken to create bathymetric maps for topography analysis, $T_{\text{loss,range}}$ is the time taken for loss calculations along with the modified propagation ranges of each sensor and $T_{\text{deployment}}$ is the time taken to deploy all sensors to achieve 100% coverage.

**Table 1 Comparative analysis of proposed method with existing methods based on coverage.**

| Method | Terrain size (m) | Comm. range (m) | Coverage (%) |
|---|---|---|---|
| Proposed method *vs. Xia et al. (2023b)* | 300 × 300 × 300 | 50 | **100% *vs* Not specified** |
| Proposed method *vs. Xia et al. (2023a)* | 120 × 120 × 120 | 60 | **100% *vs* Not specified** |
| Proposed method *vs.* Random deployment | 3,000 × 3,000 × 3,000 | 250 | **100% in both** |
| Proposed method *vs. Senel, Akkaya & Yilmaz (2013)* | 100 × 100 × 500 | 18 | **100% *vs* 90%** |
| Proposed method *vs. Du, Xia & Zheng (2014)* | 200 × 200 × 200 | 80 | **100% *vs* Not specified** |
| Proposed method *vs. Jin et al. (2018)* | 6,000 × 6,000 × 3,000 | 500 | **100% *vs* 96.82%** |

**Table 2 Comparative analysis of proposed method with existing methods based on sensors needed.**

| Method | Terrain size (m) | Comm. Range (m) | Sensors needed |
|---|---|---|---|
| Proposed method *vs. Xia et al. (2023b)* | 300 × 300 × 300 | 50 | **107 *vs* 100** |
| Proposed method *vs. Xia et al. (2023a)* | 120 × 120 × 120 | 60 | **4 *vs* 6-16** |
| Proposed method *vs.* Random deployment | 3,000 × 3,000 × 3,000 | 250 | **421 *vs* 469** |
| Proposed method *vs. Senel, Akkaya & Yilmaz (2013)* | 100 × 100 × 500 | 18 | **536 *vs* 700** |
| Proposed method *vs. Du, Xia & Zheng (2014)* | 200 × 200 × 200 | 80 | **7 *vs* 6** |
| Proposed method *vs. Jin et al. (2018)* | 6,000 × 6,000 × 3,000 | 500 | **80 *vs* 150** |

**Table 3 Comparative analysis of proposed method with existing methods based on complexity.**

| Method | Terrain size (m) | Comm. Range (m) | Complexity |
|---|---|---|---|
| Proposed method *vs. Xia et al. (2023b)* | 300 × 300 × 300 | 50 | $O(m \log n)$ ***vs.*** $O(mn)$ |
| Proposed method *vs. Xia et al. (2023a)* | 120 × 120 × 120 | 60 | $O(m \log n)$ ***vs.*** $O(N^2)$ |
| Proposed method *vs.* Random deployment | 3,000 × 3,000 × 3,000 | 250 | $O(m \log n)$ ***vs.*** $O(n^2)$ |
| Proposed method *vs. Senel, Akkaya & Yilmaz (2013)* | 100 × 100 × 500 | 18 | $O(m \log n)$ ***vs.*** $O(mn)$ |
| Proposed method *vs. Du, Xia & Zheng (2014)* | 200 × 200 × 200 | 80 | $O(m \log n)$ ***vs.*** specified as high in article |
| Proposed method *vs. Jin et al. (2018)* | 6,000 × 6,000 × 3,000 | 500 | $O(m \log n)$ ***vs.*** specified as high in article |

# RESULTS AND EVALUATION

## Comparative analysis with existing underwater deployment methods

This section presents a comparative evaluation of our proposed method against several existing underwater sensor deployment strategies. The comparison is based on coverage, number of sensors required, and computational complexity, as shown in Tables 1, 2, and 3.

To ensure a fair and consistent basis for comparison, we created simulation environments that replicate the deployment conditions reported by the respective studies. All simulations were executed using Python 3.10.11 on a notebook computer equipped with an Intel i7 12[th]-generation processor, 16 GB RAM, and a dedicated NVIDIA graphics card.

**Environmental simulation parameters:** To ensure consistency across all simulations of our proposed method, we adopted fixed underwater environmental conditions. We then compared its performance with existing state-of-the-art methods over the specific areas

and communication ranges defined in their respective literature, focusing on metrics such as coverage, number of sensors required, and computational complexity.

- **Acoustic frequency:** 14 kHz
- **Water pH:** 8.0
- **Salinity:** 35 PSU (Practical Salinity Units)
- **Temperature:** 4 °C

These parameters were used to realistically model signal attenuation and communication range, especially when the original methods did not explicitly consider transmission losses.

**Methodological classification:** To better understand the specific objectives addressed by each deployment strategy, we present a methodological classification outlining their respective goals and focus areas.

- **Target/Event-based deployment:** *Xia et al. (2023b)*, *Du, Xia & Zheng (2014)*, and *Xia et al. (2023a)* primarily focus on covering sparse event locations rather than achieving full 3D volumetric coverage.
- **Partial or incomplete volumetric coverage:** *Senel, Akkaya & Yilmaz (2013)* aim for generalized area coverage but do not guarantee completeness.
- **Volumetric coverage (3D Grid):** *Jin et al. (2018)* and our proposed method strive to cover the entire volume, though Jin et al.'s method achieves only 96.82% coverage.

**Discussion:** *Xia et al. (2023b)* and *Du, Xia & Zheng (2014)* are event-centric and do not guarantee full volume coverage. They also assume ideal, uniform communication ranges without incorporating signal loss due to environmental conditions. *Xia et al. (2023a)* employs random deployment without any coverage validation or loss modeling. The method by *Senel, Akkaya & Yilmaz (2013)* fails to ensure 100% coverage and has relatively high complexity. In contrast, our method achieves complete 3D coverage while using fewer sensors and maintaining lower computational complexity. *Jin et al. (2018)* aims for volumetric coverage but achieves only 96.82% with significantly more sensors and higher complexity.

## Deployment results over an extracted real-world RoI

**Study area: The RoI lies in the Bay of Bengal, near to the eastern coast of India.**

The extracted RoI specifically includes the coastal waters near the Indian states of Andhra Pradesh and Tamil Nadu. The RoI has a rich marine environment with abundance of flora and fauna. Therefore, this strategic location is significant for its relevance in various studies concerned with environmental monitoring. The boundaries of the RoI are determined based on the geographic coordinates-longitude, latitude and depth and are listed in Table 4.

Figure 9 shows the extracted RoI on Google Earth and 10 shows a zoomed view of it and represents the initial set of data points plotted in a 3-D space. These data points represent

**Table 4 Geographical coordinates for the real-world RoI.**

| Parameter | Minimum | Maximum |
|---|---|---|
| Latitude | 14°10′3.396″N | 14°11′36.3516″N |
| Longitude | 81°2′53.4732″E | 81°4′34.6836″E |
| Depth | 3,056 m | 3,096 m |

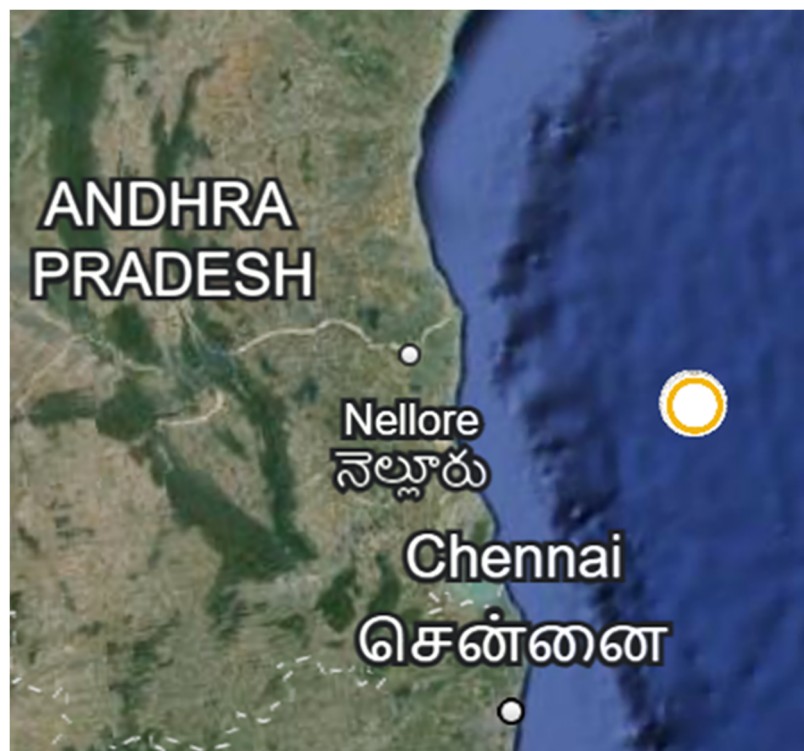

**Figure 9 Three-dimensional visualization of the extracted RoI based on geographic coordinates and depth.** This initial 3D point cloud consists of data points derived from Google Earth and processed *via* the bathymetric data extraction pipeline. Each point reflects a unique location characterized by its latitude, longitude, and depth, and forms the basis for pent. (Map data ©2025 Google).

specific features or measurements of a RoI. The coordinates (longitude, latitude, and depth) of each data point provide a comprehensive representation of the data within the region.

Table 5 shows the parameters for computation over the extracted RoI and Table 6 shows the values of the resulted outcomes.

Figure 10 presents the zoomed RoI on the map, which serves as the basis for the bathymetric analysis. Figure 11 illustrates both the 2-Dimensional and 3-Dimensional views of the bathymetric map corresponding to the extracted RoI, representing the initialization stage for subsequent acoustic modeling. The simulated 3D terrain generated from the extracted data points is depicted in Fig. 12.

Figure 13 shows the sensors deployed on the extracted ROI in both 2D and 3D formats after applying the deployment algorithm. Please note that, due to software limitations, the

**Table 5 Parameters for computations over the extracted RoI.**

| Parameter | Value |
|---|---|
| Area (sq. km) | 8.9 |
| Frequency (kHz) | 14 |
| Depth (m) | 3,056–3,096 |
| Salinity (ppt) | 35 |
| Temperature (°C) | 4 |
| pH | 8 |
| Propagation range (meters) | 250 |

**Table 6 Obtained outcomes.**

| Outcome | Value |
|---|---|
| Threshold transmission loss (dB) | 48.292581 |
| Computed transmission loss (dB) | 48.292581–48.294478 |
| Modified propagation range (m) | 249.94742–250 |
| Number of sensors needed | 522 |
| Coverage attained | 100% |

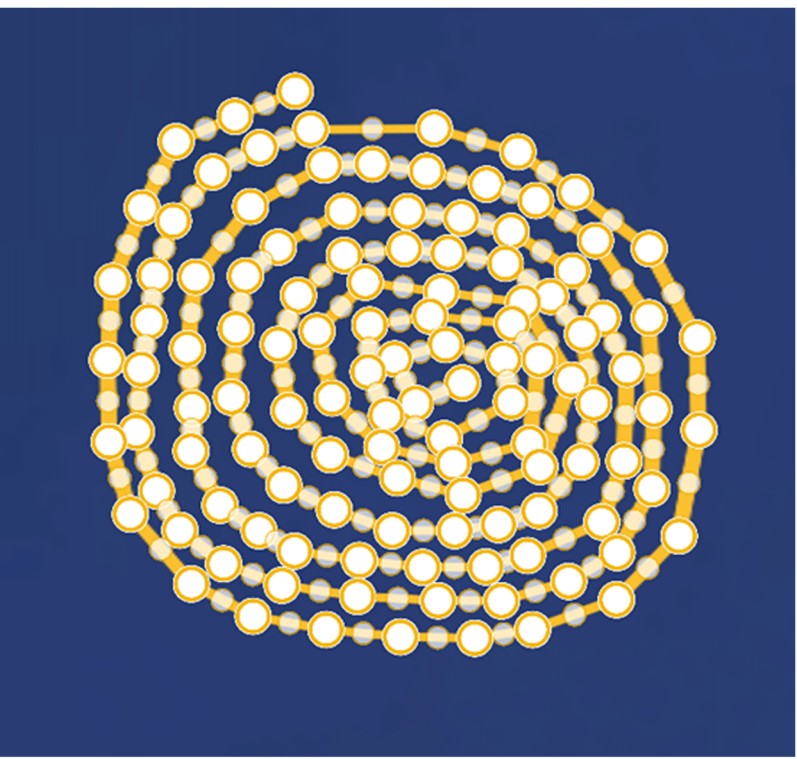

**Figure 10 Zoomed RoI on map (Map data © 2025 Google).**

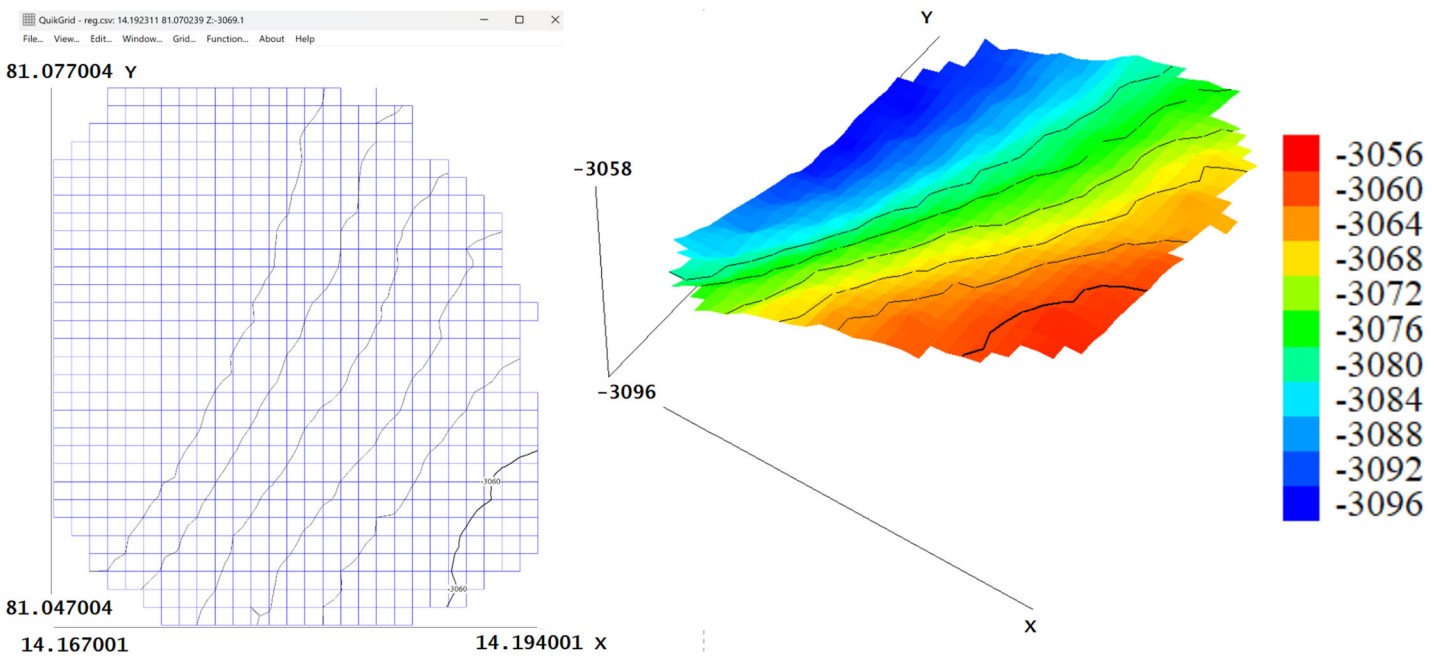

**Figure 11 2D bathymetric representation and 3D surface visualization of the bathymetric map for the selected region of interest (RoI).** The Y-axis denotes longitude (in degrees), the X-axis denotes latitude (in degrees), and in the 3D view, depth (in meters) is represented along the Z-axis.

3D view does not accurately represent distribution along depth; however, each red marker may correspond to multiple sensors placed within the region.

Figure 14 represents a 3-D simulation of the deployed sensors based on the analysis from Fig. 12. All the sensors are placed optimally to provide 100 percent coverage of the RoI. Optimal placement ensures that there are no blind spots in the region, and that all areas are properly monitored. The figure verifies that this strategy has successfully achieved complete coverage important for comprehensive monitoring. For this specific RoI, it was found that 522 sensors were required to achieve 100 percent coverage. This number is useful because it gives a standard for the resources needed to monitor the region effectively. In this specific deployment scenario, the algorithm determined that 522 sensors were required to achieve 100% coverage over the 8.9 sq. km RoI, located off the eastern coast of India in the Bay of Bengal. The relatively large area, combined with the environmental constraints outlined in Table 5—particularly the considerable depth (3,056–3,096 meters), high salinity, and cold water temperature—necessitates a dense sensor network to ensure reliable communication and data collection. The conservative propagation range limit of 250 m, computed based on threshold transmission loss, further contributes to this requirement. Additionally, the use of a greedy deployment algorithm, which prioritizes local optimization for sensor placement, may result in a higher sensor count to avoid any coverage gaps. While the number may appear high, it reflects a trade-off made to ensure complete coverage under stringent environmental and performance conditions. In practical applications, this figure can be optimized by adjusting the acceptable coverage

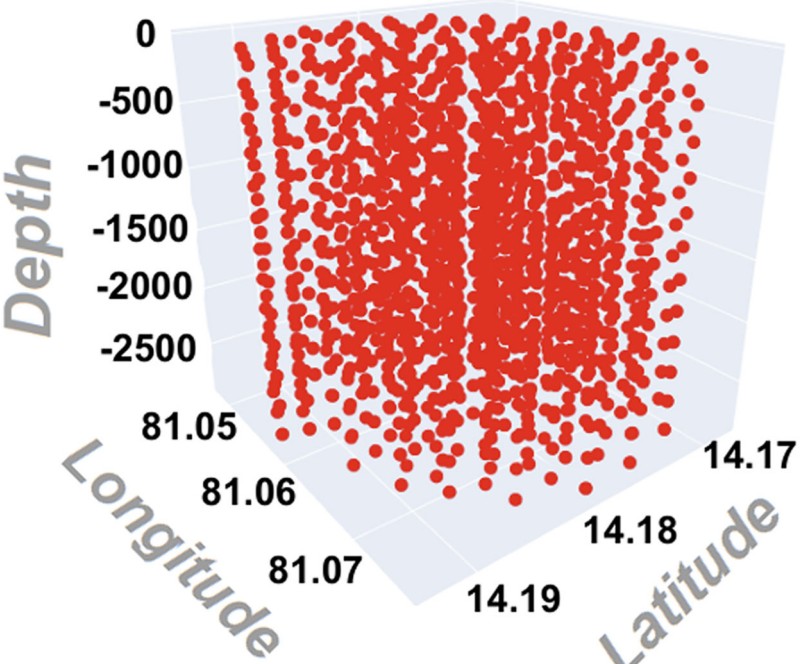

**Figure 12 3-D Simulation of extracted data points on RoI.**

threshold or relaxing propagation parameters to reduce deployment cost; however, this may result in overlapping coverage in certain areas.

Figure 15 shows the distribution of number of sensors as coverage representing the variation in coverage as we increase the number of sensors. When the coverage reaches 1, the algorithm stops and gives the total number of sensors required which are 522 in this case.

Figure 16 is a more effective demonstration which shows how depth influences signal propagation characteristics through modified propagation. The figure is a closer view, created by analyzing the simulations, where the variation of transmission losses at different depths is calculated using Eq. (4), starting from TL_TH towards greater. This implies that, as the losses increase with decrease in depth, the propagation range for effective communication between sensors is reduced.

## Computing execution time of the simulation process

The total time taken by all the phases in the simulation process can be calculated using Eq. (10). $T_{extract}$ *i.e.*, the data extraction time came out to be 89.8236 s for all data points. $T_{bathymetry}$, the time taken to create bathymetric map for topography analysis was 0.1396 s overall. $T_{loss,range}$, time taken to calculate absorption losses, transmission losses and new propagation range of each sensor was found to be 1.3804 s (being 0.003 s per sensor on

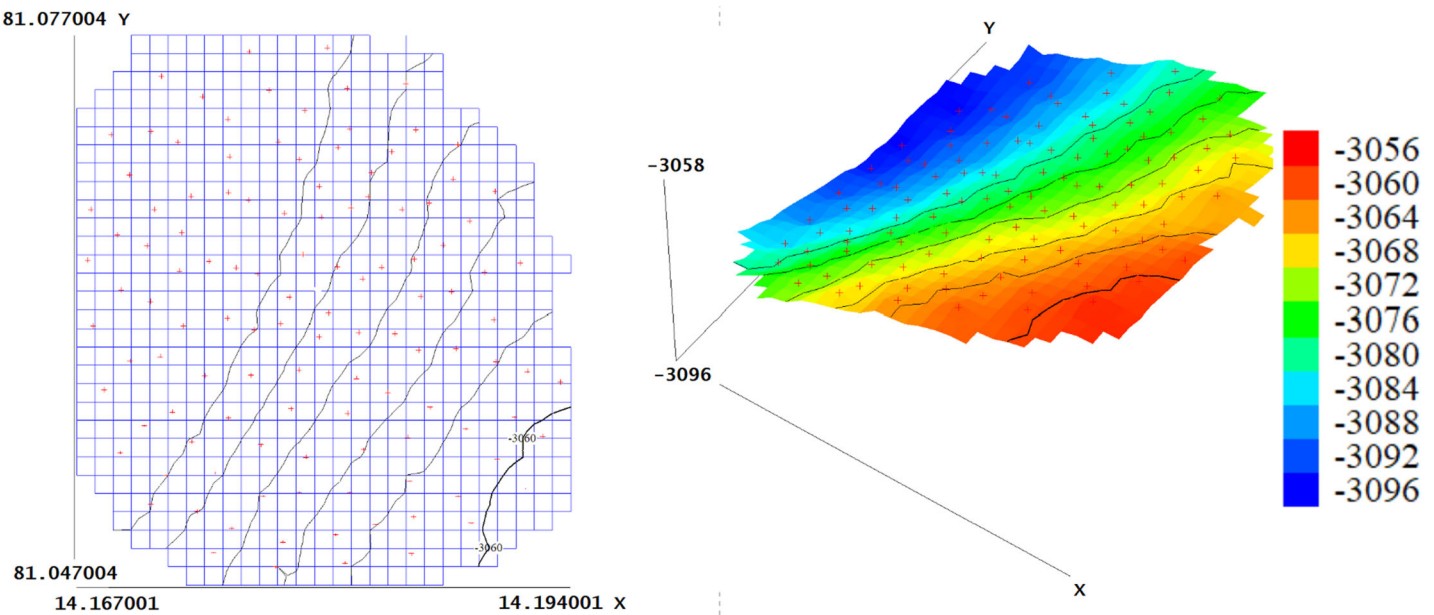

**Figure 13 2D and 3D visualizations of the bathymetric map with deployed sensor locations over the selected region of interest (RoI).** Longitude (in degrees) is plotted along the Y-axis, latitude (in degrees) along the X-axis, and depth (in meters) is shown along the Z-axis in the 3D view.

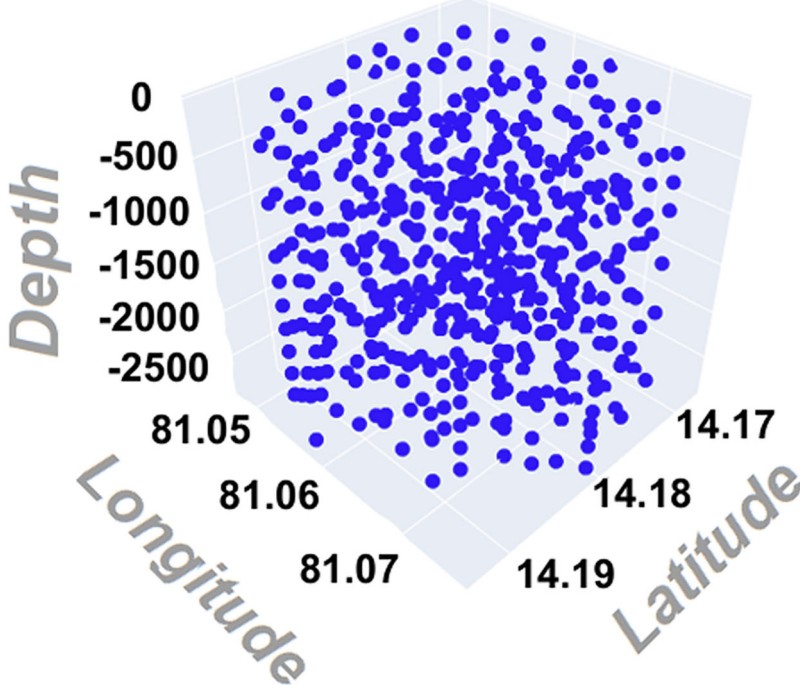

**Figure 14 3-D simulation of deployed sensors on RoI.**

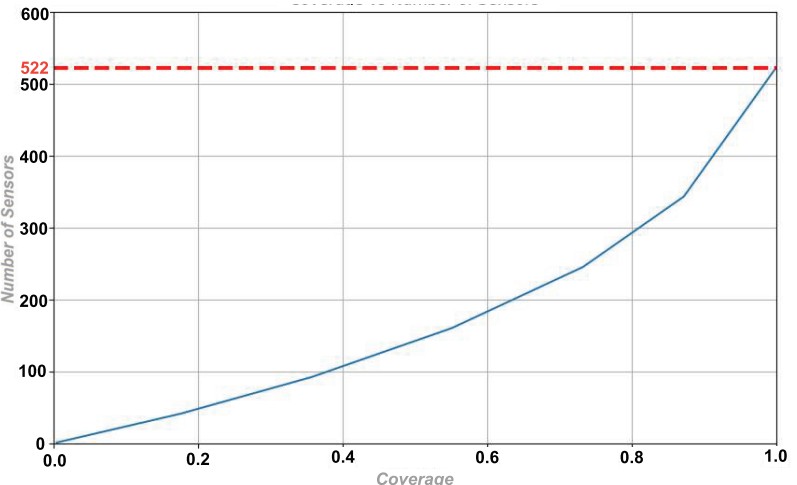

**Figure 15  Distribution of number of sensors with coverage.**

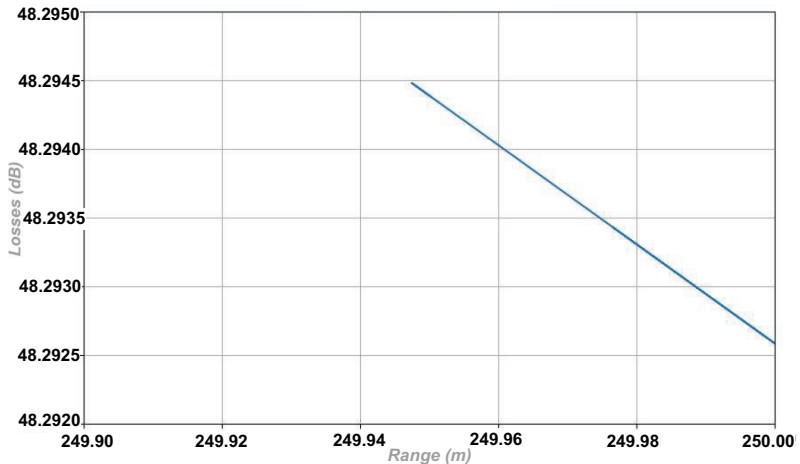

**Figure 16  Distribution of modified propagation range with losses.**

average). Finally the deployment time $T_{deployment}$ taken to achieve 100% region coverage came out to be 629.9723 s (being 1.2068 s for each sensor on average). Ultimately, the total time $T_{total}$ is calculated as 89.8236 + 0.1396 + 629.9723 = 719.9355 s. It is noteworthy that all reported values are derived from the algorithm's execution time observed during simulation runs, and may vary with changes in computational environment or simulation configurations.

Figure 17 shows the distribution of time with respect to the number of sensors deployed. It is observed that the rate of sensor deployment decreases over time *i.e.*, the time taken to deploy sensors increases with number of sensors. This behavior is expected because the algorithm does not keep track of already visited data points due to memory constraints, so it becomes more time consuming to identify new optimal sensor location each time.

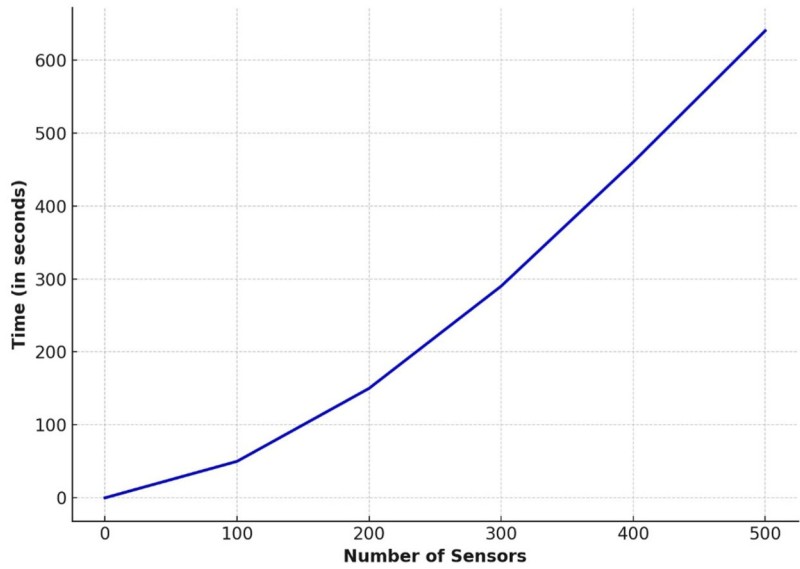

**Figure 17  Number of sensors *vs* deployment time.**   

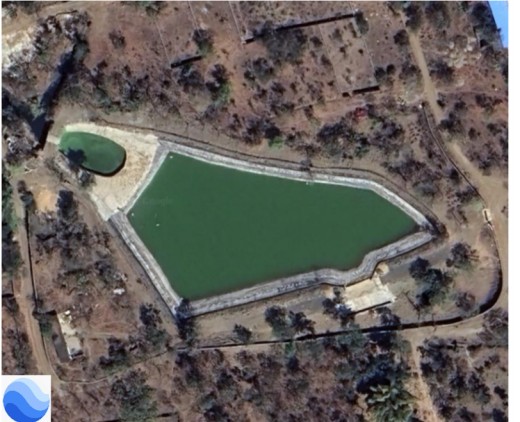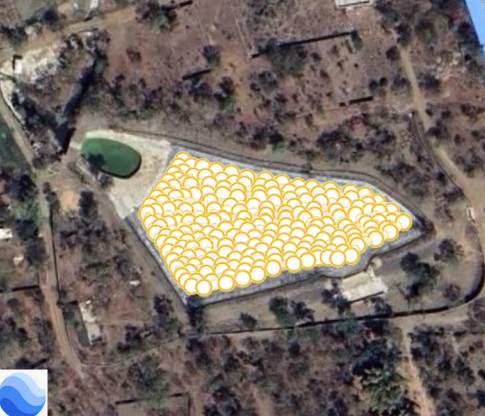

**Figure 18  Extracted RoI for real-world deployment (Map data ©2025 Google).**

## Prototypical deployment: a real-world implementation

**Study area:** An artificial lake in Simrol, Indore, Madhya Pradesh with an area of 4,260 sq. meters. It has an ideal condition for the underwater sensor network deployment and its natural setting with clean and suitable environment, makes it a perfect place for our work.

Figure 18 shows the RoI which is the entire region covered by the lake, extracted using Google Earth. Figure 19 shows the 2-Dimensional and 3-Dimensional bathymetric maps of the extracted ROI, as analyzed using QuikGrid. After implementing the sensor deployment algorithm, we figure out that 3 sensors are required to achieve 100% coverage of the extracted RoI. The underwater conditions of the Lake allowed real-world simulation of signal attenuation and transmission losses and also helped to observe the effectiveness of

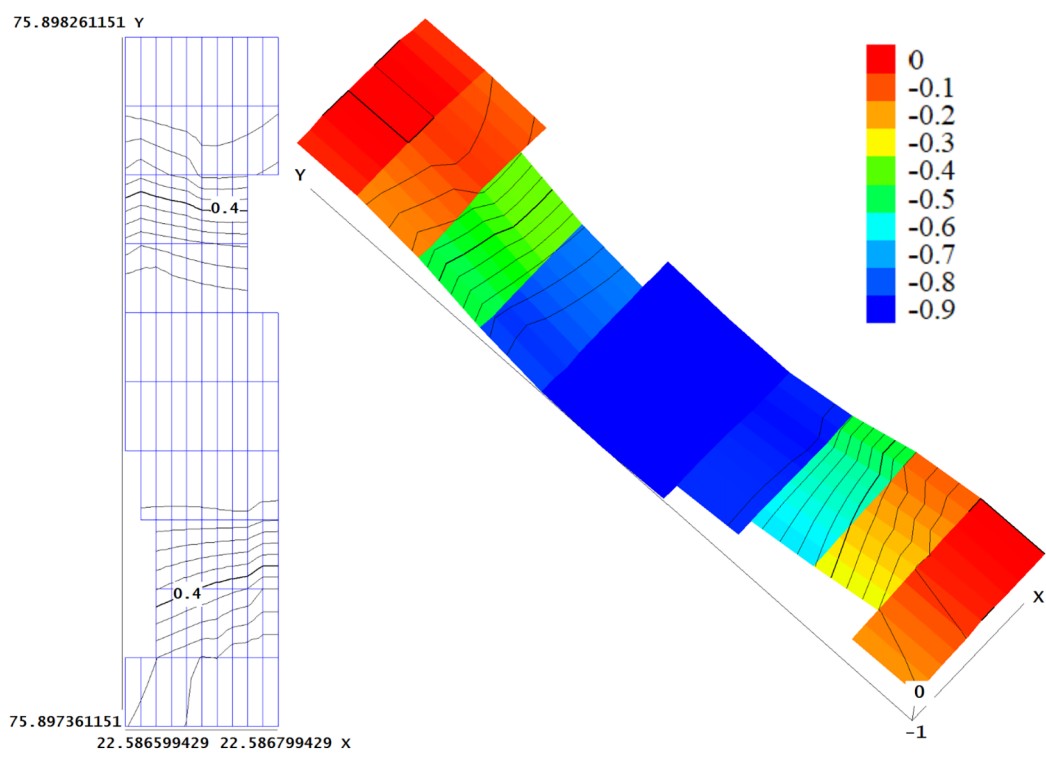

**Figure 19 2-D and 3-D bathymetry of extracted RoI.**

**Table 7 Computation results of prototypical deployment.**

| Sensor no. | Coordinates | Transmission loss (in dB) | Coverage enhancement(%) |
|---|---|---|---|
| 1 | (22.586777, 75.897762, −1.0) | 29.606588 (TL_TH) | 45.1362 |
| 2 | (22.586851, 75.898302, 0.0) | 29.606598 | 67.7043 |
| 3 | (22.586785, 75.897417, 0.0) | 29.606598 | 100.00 |

our approach. Table 7 shows the results obtained after applying the deployment algorithm on RoI. It contains the coordinates (latitude, longitude and depth) of the deployed sensors, transmission losses incurred and the coverage enhancement after placing each sensor. The modified propagation range reduced from 30.0 m to 29.999967 m which also confirms the correctness of our approach.

Figure 20 shows the bathymetric view of the locations of 3 deployed sensors in the extracted RoI.

### Equipment setup

The images in Fig. 21 shows the wireless sensor which was used in the real-time deployment. It is compact size and light weight and could conveniently connect to a smartphone and show various data like depth, temperature, bathymetry as shown in Fig. 22. The specifications are mentioned in the Table 8. Figure 23 shows a close view of a sensor placed in the Simrol Lake. The final deployed sensor positions are shown in Fig. 24 resulting in full coverage of the Simrol Lake.

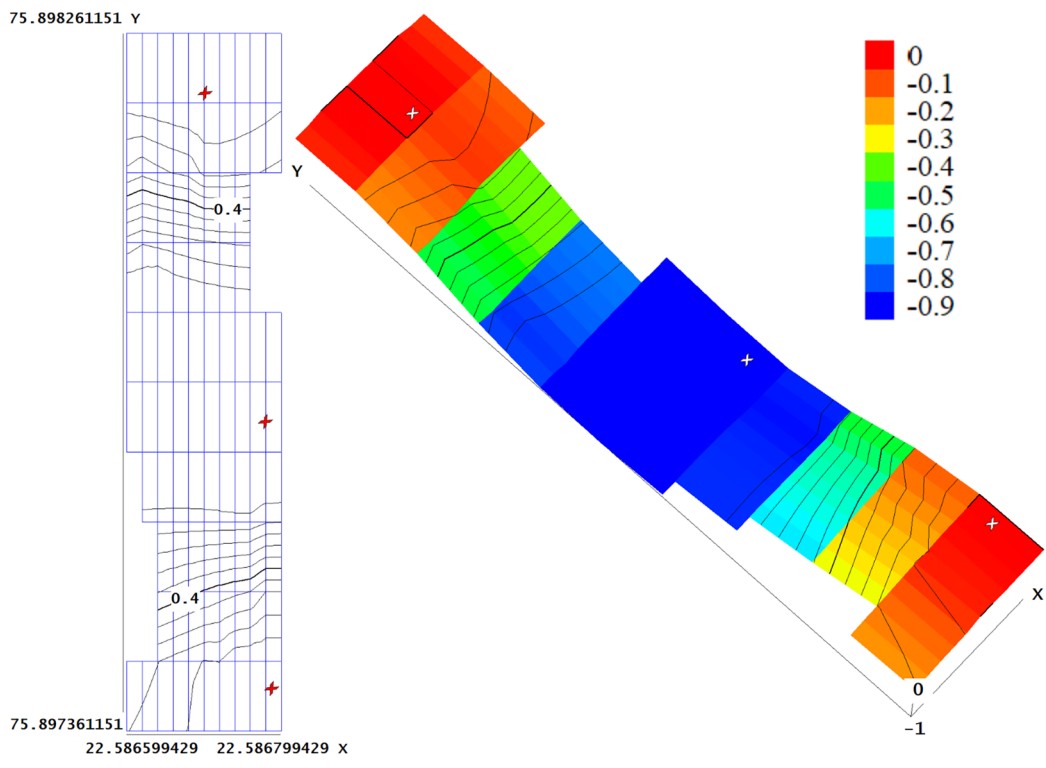

**Figure 20  2-D and 3-D bathymetry after deployment.**

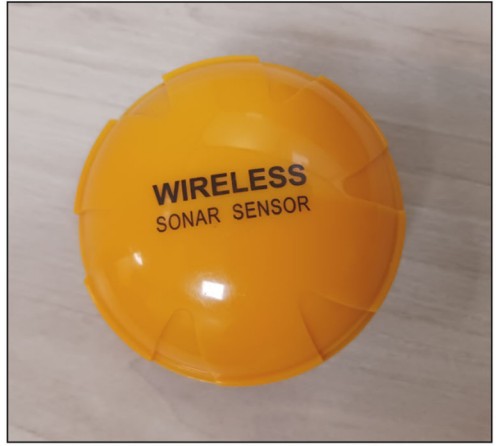
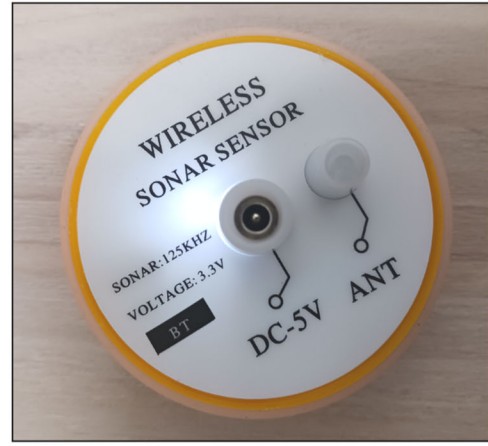

**Figure 21  Wireless sonar sensor.**                    

### Estimation of execution time for prototypical deployment

The total time taken by all the phases in the prototypical deployment process can be calculated using Eq.(10). $T_{extract}$ *i.e.*, the data extraction time came out to be 11.0448 s for all data points. $T_{bathymetry}$, the time taken to create bathymetric map for topography analysis was 0.1612 s overall. $T_{loss,range}$, time taken to calculate absorption losses, transmission losses and new propagation range of each sensor was found to be 0.4302 s

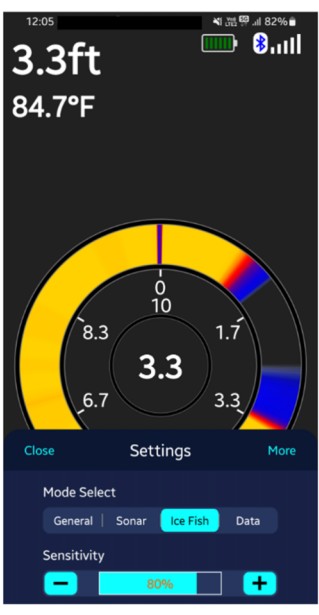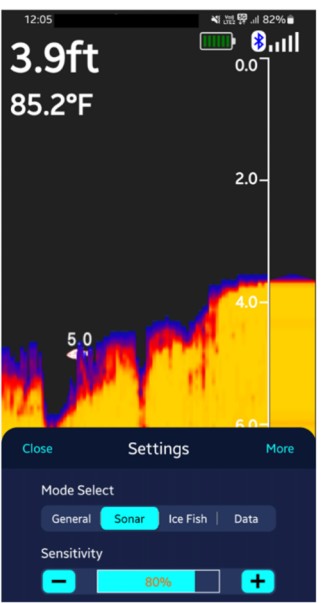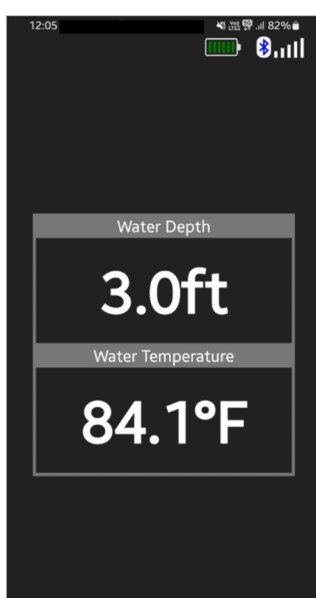

**Figure 22  Data extracted *via* sensor.**

**Table 8  Specifications of wireless sonar sensor.**

| Specification | |
|---|---|
| Depth range | 0.8–36 m |
| Transmission distance | 30 m |
| Sonar frequency | 14–125 KHz |
| Sonar radiation angle | 90° |
| Wireless frequency | 2.4 GHZ |
| Battery | 500 mAH lithium polymer battery |
| Operating temperature | 10–60 °C |

(being 0.1434 s per sensor on average). Finally, the deployment time $T_{deployment}$ taken to achieve 100% region coverage came out to be 1.2 s (being 0.4 s for each sensor on average). Ultimately, the total time $T_{total}$ is calculated as 11.0448 + 0.1612 + 1.2 = 12.406 s.

## Deployment reliability and data-driven limitations

While the proposed greedy deployment algorithm is effective in achieving full coverage without overlap or redundancy, certain limitations may arise due to dependencies on external data quality. The accuracy and resolution of the underlying bathymetric dataset play a critical role in the overall deployment outcome. Minor inaccuracies or undetected topographic features—such as small ridges, trenches, or abrupt depth variations—may not be captured during the data extraction process, potentially leading to localized undercoverage or slight deviations in sensor placement.

Although the algorithm is designed to incorporate spatial separation logic and propagation constraints to minimize overlap, real-world environmental uncertainties or measurement

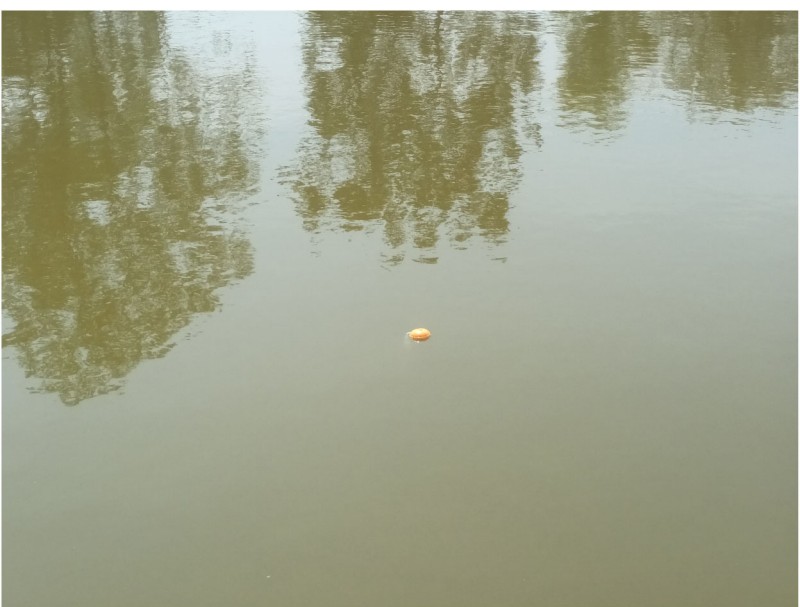

**Figure 23 Sensor placed in water (close view).**

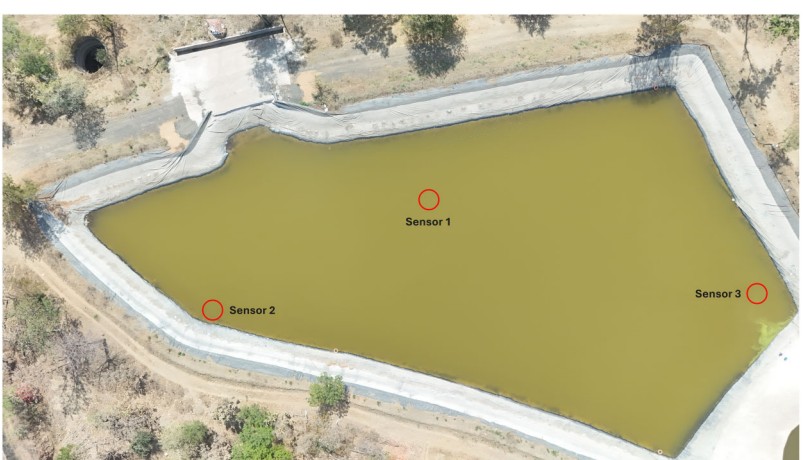

**Figure 24 Positions of sensors on Simrol Lake (Photo credit: Rupendra Singh).**

errors can influence the accuracy of the computed sensor positions. It is important to note that these limitations are not inherent to the algorithm itself but are instead a result of practical constraints associated with terrain modeling and data fidelity. Future enhancements may include adaptive correction mechanisms or integration of real-time terrain feedback to improve deployment robustness in dynamic underwater environments.

## CONCLUSION AND FUTURE WORK

UASNs face numerous challenges, including optimizing coverage efficiency, managing computational complexity, and addressing environmental factors such as salinity, depth,

water temperature, signal frequency, and signal attenuation that impact sensor performance. Our proposed approach addresses these challenges through a novel greedy method for optimal sensor deployment, which maximizes coverage using a minimal number of sensors while accounting for transmission losses in dynamic underwater environments. We utilized satellite imagery to extract the RoI and constructed detailed bathymetric maps to analyze the underwater topography. Simulations were performed that included the estimation of transmission losses under various environmental constraints, which significantly enhanced the accuracy of sensor placement and coverage estimation. Comparative analysis with previous studies demonstrated the superior performance of our approach in terms of both sensor placement and coverage efficiency. Real-world prototypical implementations validated the practicality and effectiveness of our method by calculating the optimal number of sensors required to achieve significant coverage.

Future work may focus on enhancing failure management mechanisms to minimize the need for frequent maintenance and to improve the long-term reliability and durability of underwater sensor networks. Additionally, the current study can be extended to incorporate multipath effects, which significantly influence acoustic signal propagation. We aim to address these aspects by integrating adaptive reconfiguration mechanisms for post-deployment updates and failure recovery, thereby enhancing network resilience and operational performance in real-world underwater environments.

### Funding
This work was supported by the Research Council of Norway through the INTPART DTRF project. The funders had no role in study design, data collection and analysis, decision to publish, or preparation of the manuscript.

### Grant Disclosures
The following grant information was disclosed by the authors:
INTPART DTRF Project.

### Competing Interests
The authors declare that they have no competing interests.

### Author Contributions
- Shekhar Tyagi conceived and designed the experiments, performed the experiments, analyzed the data, performed the computation work, prepared figures and/or tables, authored or reviewed drafts of the article, and approved the final draft.
- Akshat Shah conceived and designed the experiments, performed the experiments, analyzed the data, performed the computation work, prepared figures and/or tables, authored or reviewed drafts of the article, and approved the final draft.
- Abhishek Srivastava conceived and designed the experiments, performed the experiments, analyzed the data, performed the computation work, prepared figures and/or tables, authored or reviewed drafts of the article, and approved the final draft.

## Data Availability

The code is available at GitHub and Zenodo:

- https://github.com/akshat-shah-2003/Underwater-Sensor-Project.

- akshat-shah-2003. (2025). akshat-shah-2003/Underwater-Sensor-Project: v1.0–Initial Release (v1.0). Zenodo. https://doi.org/10.5281/zenodo.17264448.

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
