# Peer review of "Environment aware greedy deployment strategy for underwater acoustic sensor networks using bathymetric mapping and transmission loss modeling"

_PeerJ Computer Science, doi:10.7717/peerj-cs.3321_

## Round 0.1 · original submission · Major Revisions

· Academic Editor

Major Revisions

Reviewer 1 ·

Basic reporting

There are multiple typos and grammatical errors throughout the manuscript that need to be addressed.

Wikipedia is not a peer-reviewed source and should not be cited within the manuscript. The use of Wikipedia citations on lines 191 and 376 should be replaced or removed.

Several acronyms are defined multiple times in the manuscript (e.g. UASN and ROI) please define these once in the manuscript (not including the abstract). Please define AUV and ROV on Line 199. Please define UWSN earlier in the manuscript.

Lines 486-488 are confusing with the curve in Figure 17, please flip the x and y axes. It also seems that the time to deploy >100 sensors increases significantly. How is this curve computed, and why is there a large increase around 100 sensors? How is T deployment estimated?

There are multiple typos and grammatical errors throughout the manuscript. For brevity, I have noted a few locations below. Line 197: “(ROI)” should be removed; Line 201: is a grammatical error; Line 212: “can termed”; Line 226 “and” should be removed; Line 244: “compute” should be “computes”; Lines 279-280: remove the part of the sentence “The methodology aims to maximize… “; Line 285: remove “in”; Line 287: “compare” should be “compares”; Line 288: remove “on”; Line 353: remove “so”; Lines 399-400: grammatical error; Line 431: “increased up” should be “increased”; Line 445: “is having” should be “has”; Line 486: the sentence is confusing.

It is difficult to see the positions of the deployed sensors in Figure 13. In the 3D case, are only the positions at depth considered?

Indicate units in Figures 4, 5, 11, and 13.

Equations 4 and 5 are the same, please remove equation 5. Moreover, Lines 238-243 are already stated in the previous section and should be removed.

Please indicate what units are the propagation range on Line 231.

Experimental design

Although the idea of using transmission loss as a requirement for sensor placement using the greedy algorithm is interesting, the model doesn’t factor in multipath from shallow environments, bathymetry, or variations of environmental factors (e.g. temperature at depth). The manuscript does not detail steps to update positions after deployment, which would be beneficial for readers.

Please indicate the TL threshold considered for the real-world implementation.

Validity of the findings

Please describe the comparative analysis in Tables 1-3 more, it is unclear how the proposed method compares when each method follows a specific approach (i.e., which ones are volumetric coverage vs target). It is also unclear what frequencies, pH, salinity, and temperature are being used for each example case.

Please describe where Advanced Converter is getting bathymetry data from and what the resolution is.

Lines 458-459, refer to Figure 12 and state: “This figure is necessary to understand how depth affects the propagation range of signals or measurements,” but does not show how depth impacts TL. This is only an initialization, and the plot in Figure 16 reflects this sentence better.

Please provide information on how Figure 16 is calculated.

Reviewer 2 ·

Basic reporting

The abstract provides a good overview of the study; however, it is somewhat dense. Simplifying complex sentences and explicitly quantifying the reported improvements would enhance clarity.
While key challenges such as signal attenuation and salinity are acknowledged, the manuscript would benefit from a clearer discussion of their quantitative impact on network performance.
The manuscript provides a general motivation, but the specific research gap, particularly in relation to environmental modeling and non-ideal underwater conditions, is not clearly defined. A clearer articulation would better highlight the novelty.
The related work section reviews a variety of techniques, but it would be more informative if it also included a critical assessment of the limitations of these approaches.
Several recent and thematically relevant works published in 2023–2025 are missing and could enhance the contextual depth of the literature review. For example, the authors may consider reviewing the following contributions.

-Self-Organized Proactive Routing Protocol for NON-UNIFORMLY DEPLOYED Underwater Networks.
-Reliable, Energy-Optimized, and Void-Aware (REOVA) Routing Protocol with STRATEGIC DEPLOYMENT in Mobile Underwater Acoustic Communications.
-GRID DEPLOYMENT SCHEME for Enhancing Network Performance in Underwater Acoustic Sensor Networks
-POWER-CONTROL-BASED ENERGY-EFFICIENT DEPLOYMENT for Underwater Wireless Sensor Networks with Asymmetric Links
Figures 7 to 9 are useful but would benefit from more descriptive captions to guide the reader and enhance interpretability.
Certain technical terms, such as “threshold state” and “greedy deployment,” are introduced without definition. These should be clearly explained when first mentioned to support reader comprehension.

Experimental design

The reliance on external tools such as Google Earth and online converters for RoI extraction is practical, but the manuscript should address potential reproducibility concerns and describe any associated limitations.
The methodology does not discuss the potential error margins or accuracy limitations of the extracted bathymetric data. Including such a discussion would improve methodological transparency.
The assumptions regarding environmental parameters like turbidity and pH at maximum depth require further justification to confirm their scientific basis.
The assumptions regarding environmental parameters like turbidity and pH at maximum depth require further justification to confirm their scientific basis. Additionally, statement like “because transmission loss is least at the maximum depth being inversely proportional to depth” should be cite a specific published source.
Using the maximum depth to determine sensor range thresholds is a key assumption in the methodology; a clearer rationale or empirical backing would strengthen its validity.

Validity of the findings

In some cases, the number of deployed sensors (e.g., 522) seems high. A brief analysis of cost-performance tradeoffs or deployment efficiency would help contextualize these figures.
The claim of achieving “100% coverage” is promising, but could be overly optimistic without addressing potential factors such as coverage overlap, redundancy, or practical deployment challenges.
The results section describes performance improvements, but lacks a deeper analysis of why and how the proposed method outperforms others. A more critical discussion would enhance the impact of the findings.
The manuscript does not explicitly discuss the limitations of the proposed approach. A transparent account of both methodological and deployment-related constraints would add credibility to the study.

Additional comments

Title: “Environment Aware Greedy Deployment Strategy for Underwater Acoustic Sensor Networks Using Bathymetric Mapping and Transmission Loss Modeling”.

This research work addresses a meaningful and timely problem in underwater sensor deployment using environmentally aware models. The integration of bathymetric data, loss modeling, and greedy coverage estimation is sound. The proposed method shows promise, particularly in integrating environmental factors, however, several aspects particularly reproducibility, validation scale, statistical robustness, and novelty justification require clearer elaboration.

Annotated reviews are not available for download in order to protect the identity of reviewers who chose to remain anonymous.

---

## Round 0.2 · accepted · Accept

· Academic Editor

Accept

The reviewer has no further concerns.

Reviewer 2 ·

Basic reporting

The revised manuscript demonstrates clear and professional writing, a well-structured format, and effective use of figures and tables. The authors have complied with previously suggested revisions, and the inclusion of raw data enhances transparency and reliability.

Experimental design

The manuscript presents original primary research that falls well within the journal’s aims and scope. It reflects a rigorous investigation conducted to a high technical and ethical standard.

Validity of the findings

All foundational data are supplied, reliable, analytically rigorous, and carefully validated, reinforcing the study’s credibility while emphasizing the originality of the research.

Additional comments

The study is well-conducted and holds strong potential to benefit future research. While it opens several avenues requiring further exploration, the current work is commendable and merits acceptance for publication.